# Bacteria can maintain rRNA operons solely on plasmids for hundreds of millions of years

Mizue Anda [1,2] ✉, Shun Yamanouchi[2], Salvatore Cosentino [1,2], Mitsuo Sakamoto [3], Moriya Ohkuma[3], Masako Takashima [3], Atsushi Toyoda [4] & Wataru Iwasaki [1,2,5,6,7,8] ✉

It is generally assumed that all bacteria must have at least one rRNA operon (*rrn* operon) on the chromosome, but some strains of the genera *Aureimonas* and *Oecophyllibacter* carry their sole *rrn* operon on a plasmid. However, other related strains and species have chromosomal *rrn* loci, suggesting that the exclusive presence of *rrn* operons on a plasmid is rare and unlikely to be stably maintained over long evolutionary periods. Here, we report the results of a systematic search for additional bacteria without chromosomal *rrn* operons. We find that at least four bacterial clades in the phyla Bacteroidota, Spirochaetota, and Pseudomonadota (Proteobacteria) lost chromosomal *rrn* operons independently. Remarkably, Persicobacteraceae have apparently maintained this peculiar genome organization for hundreds of millions of years. In our study, all the *rrn*-carrying plasmids in bacteria lacking chromosomal *rrn* loci possess replication initiator genes of the Rep_3 family. Furthermore, the lack of chromosomal *rrn* operons is associated with differences in copy numbers of *rrn* operons, plasmids, and chromosomal tRNA genes. Thus, our findings indicate that the absence of *rrn* loci in bacterial chromosomes can be stably maintained over long evolutionary periods.

Bacterial chromosomes are traditionally distinguished from plasmids by the fact that chromosomes encode essential genes[1,2]. The canonical essential genes include 16S, 23S, and 5S rRNA genes, which usually constitute an rRNA operon (*rrn* operon). All bacteria had been believed to have at least one *rrn* operon on their chromosomes, even if additional *rrn* operons could be found on extrachromosomal replicons[3–8]. Theoretical and experimental studies suggest that a population consisting of individuals with sole *rrn* operon only on the plasmid are unstable[9] and a second copy of essential genes cannot be maintained stably on plasmids[10], respectively.

This widespread belief was challenged by the unexpected finding of a plant-associated bacterium *Aureimonas ureilytica* (family Aurantimonadaceae, order Rhizobiales, class Alphaproteobacteria), which carries its sole *rrn* operon on a high copy number plasmid without partition system genes[11]. This finding attracted considerable attention in terms of the diversity and evolution of bacterial genomes and plasmids[12–17]. However, whether *A. ureilytica* is a unique, exceptional species and if there are essential prerequisites that allow *rrn* operons to transfer from chromosomes to plasmids have not been resolved. In addition, because other *Aureimonas* species have been revealed to

[1]Department of Integrated Biosciences, Graduate School of Frontier Sciences, the University of Tokyo, Kashiwa, Chiba 277-0882, Japan. [2]Department of Biological Sciences, Graduate School of Science, the University of Tokyo, Bunkyo-ku, Tokyo 113-0032, Japan. [3]Microbe Division/Japan Collection of Microorganisms, RIKEN BioResource Research Center, Tsukuba, Ibaraki 305-0074, Japan. [4]Advanced Genomics Center, National Institute of Genetics, Mishima, Shizuoka 411-8540, Japan. [5]Department of Computational Biology and Medical Sciences, Graduate School of Frontier Sciences, the University of Tokyo, Kashiwa, Chiba 277-0882, Japan. [6]Atmosphere and Ocean Research Institute, the University of Tokyo, Kashiwa, Chiba 277-0882, Japan. [7]Institute for Quantitative Biosciences, the University of Tokyo, Bunkyo-ku, Tokyo 113-0032, Japan. [8]Collaborative Research Institute for Innovative Microbiology, the University of Tokyo, Bunkyo-ku, Tokyo 113-0032, Japan. ✉e-mail: anda@k.u-tokyo.ac.jp; iwasaki@k.u-tokyo.ac.jp

have chromosomal *rrn* operons, how long bacteria without chromosomal *rrn* operons can avoid extinction over evolutionary timescales remains unclear. Regarding the uniqueness of *A. ureilytica*, a recent study incidentally found that an ant-associated bacteria *Oecophyllibacter saccharovorans* (family Acetobacteraceae, order Rhodospirillales, class Alphaproteobacteria) carries its sole *rrn* operon on a plasmid[18]. We thus envisioned a systematic study to identify more bacterial species without chromosomal *rrn* operons to provide clues to those fundamental questions.

In this study, by combining bioinformatic analysis and genome sequencing, we investigated if bacteria without chromosomal *rrn* operons evolved repeatedly and could be maintained for an evolutionarily long term. We found that bacteria without chromosomal *rrn* operons independently evolved at least four times in three phyla. Most notably, the family Persicobacteraceae was revealed to have lost its chromosomal *rrn* operons likely >492 million years ago (MYA), demonstrating that bacteria without chromosomal *rrn* operons can avoid extinction over geological timescales.

## Results

### Bioinformatic analysis found multiple bacteria that likely lost chromosomal *rrn* operons

To search for bacterial genomes whose *rrn* operons reside exclusively on plasmids, we downloaded 86,822 genomes from NCBI RefSeq[19]. This dataset contained numerous (90.7%) draft genomes, whose contigs were classified as neither chromosomes nor plasmids. One signature of plasmid contigs is the presence of a *rep* gene, which encodes the plasmid replication initiator Rep protein, as in the case of *rrn* plasmids of *A. ureilytica*[11]. Thus, we selected genomes whose annotated *rrn* operons were exclusively on contigs that encoded *rep* genes, using tBLASTn searches with 156 Rep protein sequences[20]. To enrich genomes that have *rrn* operons only on plasmids, we excluded genomes if one of their *rrn* operons was on a contig longer than 35 kb or encoded essential single-copy genes (Supplementary Fig. 1). We confirmed that our method made neither false-positive nor false-negative using the *A. ureilytica* genomes[11,21] and RefSeq complete genomes. Aside from the *A. ureilytica* genomes, our analysis identified three additional genomes that potentially have *rrn* operons exclusively on plasmids with *rep* genes: genomes of *Persicobacter* spp. JZB09 and CCB-QB2 (family Persicobacteraceae; order Cytophagales, class Cytophagia, phylum Bacteroidota), whose 16S rRNA genes were 99% identical to those of *Persicobacter diffluens*; and *Treponema saccharophilum* DSM2985 (family Spirochaetaceae; order Spirochaetales, class Spirochaetia, phylum Spirochaetota).

*Persicobacter* spp. JZB09 and CCB-QB2 genomes (GCF_001308105.1 and GCF_001274635.1) were sequenced using PacBio RSII, and their assembly statuses were complete and scaffold, respectively[22]. Both genomes contained three *rrn* operons, which were arranged in tandem on a single ~30-kb contig with *rep* genes (Fig. 1a). Those *Persicobacter* genomes were of particular interest because genomes of related species in the same Persicobacteraceae family have not been sequenced yet, and thus the loss of chromosomal *rrn* operons might be widely conserved across related species.

The *T. saccharophilum* genome (GCF_000255555.1) was sequenced by Genome Analyzer II and 454 GS FLX. Its rRNA genes were found only at both ends of a single 8.4-kb contig, suggesting that its *rrn* operon is present only on this circular plasmid. Other *Treponema* genomes whose assembly statuses were complete or chromosome had chromosomal *rrn* operons. Thus, *T. saccharophilum* would have recently lost its chromosomal *rrn* operons around the time of speciation.

### De novo genome sequencing revealed that bacteria lost chromosomal *rrn* operons at least four times independently

To confirm that the genomes of *P. diffluens* and *T. saccharophilum* genuinely have no *rrn* operons in their chromosomes, we obtained

their type strains (*P. diffluens* NBRC 15940^T and *T. saccharophilum* JCM 32279^T) and experimentally determined their genomes through hybrid assembly using PacBio RSII and HiSeq sequencers (Table 1 and Supplementary Table 1). Neither of those genomes had *rrn* operons in their chromosomes. Like the two complete/scaffold *Persicobacter* genomes, the genome of *P. diffluens* was confirmed to have three *rrn* operons that are arranged in tandem on a single ~30-kb plasmid harboring *rep* genes (Fig. 1a). In addition, the *T. saccharophilum* genome was confirmed to have an *rrn* operon on a single 8.4-kb plasmid with a *rep* gene (Fig. 1b). Therefore, we concluded that *P. diffluens* and *T. saccharophilum* are species that lost *rrn* operons from their chromosomes. In other words, the bacterial domain lost chromosomal *rrn* operons at least four times independently in the Bacteroidota, Spirochaetota, and Pseudomonadota phyla (i.e., *A. ureilytica* (Fig. 1c) and *O. saccharovorans* (Fig. 1d)).

Next, we determined genomes of species related to *P. diffluens* in family Persicobacteraceae: *P. pyschrovividus*, *Aureibacter tunicatorum*, and *Fulvitalea axinellae* (Table 1 and Supplementary Table 1). Their complete genomes revealed that they do not have *rrn* operons in their chromosomes but have two *rrn* operons in tandem on their smallest plasmid (13.7–25.9 kb) (Fig. 1a). Moreover, the *rrn* operons of all the six Persicobacteraceae genomes shared a common structure of *rrs* (16S rRNA)-*trnI* (tRNA^Ile)-*trna* (tRNA^Ala)-*rrl* (23S rRNA)-*rrf* (5S rRNA) except for one operon on the *Persicobacter* genomes, strongly suggesting that they share a single evolutionary origin (Fig. 1a, e). This is the first discovery that *rrn* operons are absent from bacterial chromosomes at the taxonomic family level.

### Persicobacteraceae lost their chromosomal *rrn* operons hundreds of millions of years ago

The plasmid *rrn* operons of the Persicobacteraceae species and *T. saccharophilum* were probably transferred from their chromosomal *rrn* operons or through horizontal gene transfers from distant species. To determine the evolutionary origins of their *rrn* operons, we reconstructed and compared the rRNA gene trees and genomic trees based on single-copy protein-coding genes (Fig. 1f, g and Supplementary Data 1, 2). In both cases, the two trees exhibited topological similarities (Supplementary Data 3), suggesting that their plasmid *rrn* operons were transferred from their chromosomes and not horizontally transferred from distant clades. The plasmid *rrn* operons of *A. ureilytica* and *O. saccharovorans* were also transferred from the chromosomes[11,23].

The chromosomal origins of the plasmid *rrn* operons allow us to estimate the timings of the transfer of their *rrn* operons to plasmids and their loss from chromosomes using genomic data. The genomic phylogenomic trees of the four clades were obtained from GTDB or inferred using GTDB-Tk[24,25], and their divergence times were estimated using RelTime[26] (Fig. 2a–d, Supplementary Fig. 2, Supplementary Data 4). The chromosomal *rrn* operons were estimated to have been lost 492-651 MYA, 0-221 MYA, 69.3-166 MYA, and 1.27-91.2 MYA in the ancestors of Persicobacteraceae species, *T. saccharophilum*, *A. ureilytica*, and *O. saccharovorans*, respectively, under the assumption that the chromosomal *rrn* operons were lost at the common ancestor of each clade (see Discussion). The outstanding ancient origin of the Persicobacteraceae ancestor was particularly notable for two reasons: First, it challenges the widely accepted belief that essential genes cannot be maintained stably on plasmids for an evolutionarily long term[9]. Second, comparing bacteria without chromosomal *rrn* operons between the Persicobacteraceae family and the other three clades will enable us to investigate how *rrn* operons transferred from chromosomes to plasmids and subsequently plasmids matured in the long term.

### *rrn* plasmids of bacteria without chromosomal *rrn* operons always had Rep_3-family genes

As mentioned above, we used plasmid replication initiator *rep* genes as a signature of plasmids. The 156 representative Rep proteins[20]

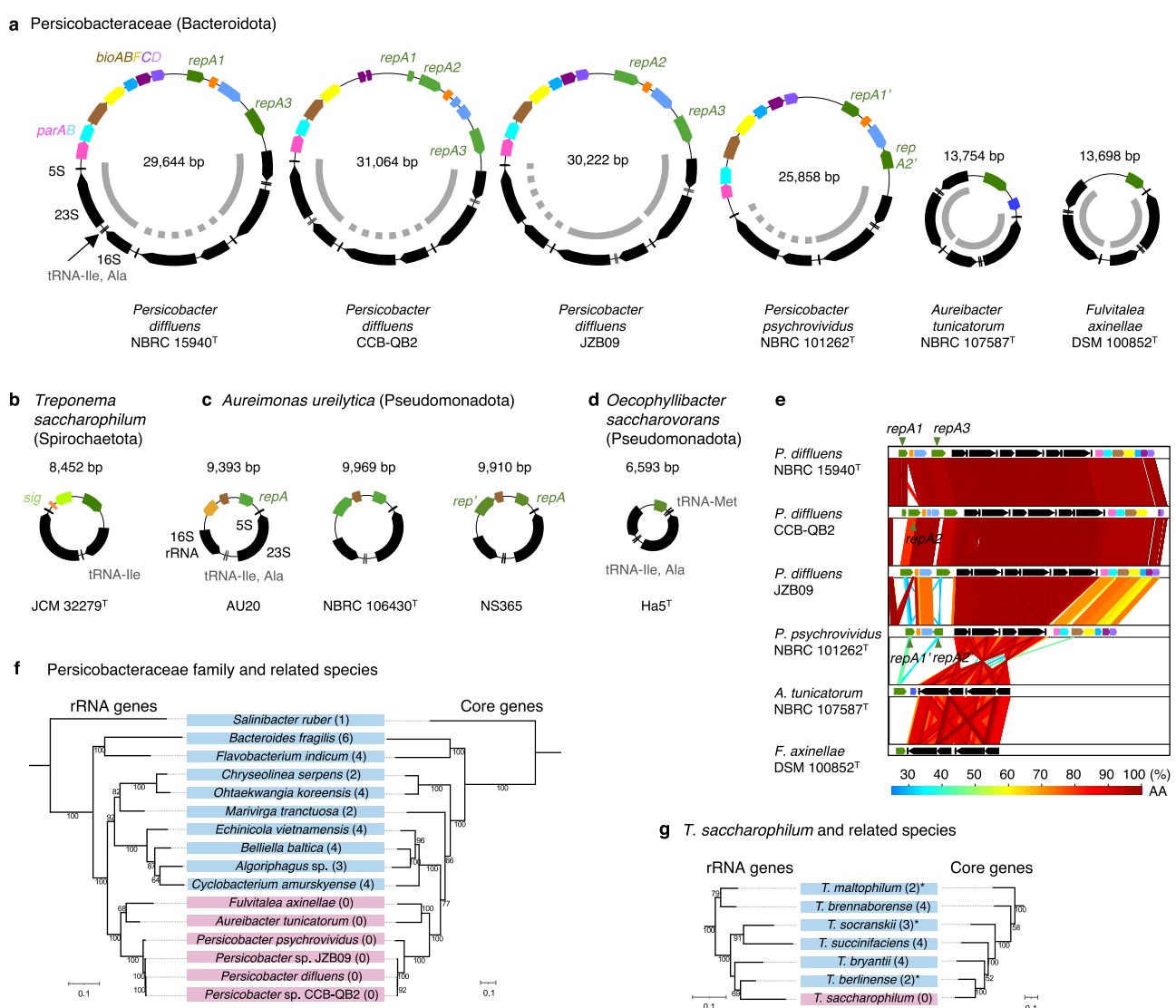

**Fig. 1 | Four bacterial clades that independently lost chromosomal *rrn* operons. a–d** Maps of the *rrn* plasmids of **a** Persicobacteraceae species, **b** *Treponema saccharophilum*, **c** *Aureimonas ureilytica*[11], and **d** *Oecophyllibacter saccharovorans*[18]. Solid and dotted inner arcs show *rrn* operon with or without tRNA genes, respectively. rRNA and tRNA genes are shown by black arrows. *rep*, *parAB*, *bioABFCD,* and *sig* genes encode plasmid replication initiators, partition-ing proteins, biotin synthetic proteins, and sigma factors, respectively. Genes whose products belong the same function are presented in the same colors. Note that *repA1A2A3* of *Persicobacter diffluens* and *repA1'A2'* of *Persicobacter psychro-vividus* have no sequence similarity. **e** Comparative synteny map of *rrn* plasmids of the Persicobacteraceae species. **f, g** Phylogenetic trees of rRNA genes and conserved single-copy protein-coding genes of **f** Persicobacteraceae species and **g** *T. saccharophilum* and their related species. Pink indicates bacteria without chromosomal *rrn* operons. Light blue indicates bacteria with chromosomal *rrn* operons. Numbers in parentheses are numbers of chromosomal *rrn* operons. Asterisks indicate genomes whose assembly levels are scaffold. Bootstrap values of >50% are shown. Scale bars indicate substitution numbers per site. See Sup-plementary Data 1–3 for strain names, conserved single-copy protein-coding genes, and results of topological test, respectively.

contained 17 families in Pfam[27] whose primary function is plasmid replication (Supplementary Data 5).

We found that all *rrn* plasmids of the four bacterial clades had Rep_3-family genes (Fig. 1a–d). Furthermore, we inferred all gene gain (birth, horizontal gene transfer, or duplication) and loss events in the ancestors that lost chromosomal *rrn* operons using PastML[28]. This analysis showed that no genes other than Rep_3-family genes were consistently gained or lost in the ancestors of the four clades (Fig. 2e). Evolutionary analysis using CAFE[29] did not identify additional gene-number expansion and contraction events common to those four ancestors. These lines of evi-dence strongly suggested that Rep_3-type plasmids represent essential prerequisites behind the losses of chromosomal *rrn* operons.

Phylogenetic analysis of the Rep_3-family genes estimated that those on several *rrn* plasmids were horizontally transferred from distant clades (such as different classes) to their ancestors (Supplementary Fig. 3). We also noted that related species of the four clades were not frequent hosts of Rep_3-family genes (Fisher's exact test $p < 0.05$, Fig. 2f). Thus, we assume that transfer of *rrn* operons from chromosomes to plasmids would have enabled those ancestors to maintain Rep_3-type plasmids, possibly because of the essentiality of the *rrn* operons. Nota-bly, in the Persicobacteraceae, we found cases that Rep_3 transferred between the *rrn* plasmid and other plasmids (open stars in Supple-mentary Fig. 3a, c, e, Fig. 1e). Moreover, the Rep_3 family was observed in all plasmids mainly as clades (Fig. 2g, diamond symbols in Supplemen-tary Fig. 3a–d), while no other Rep families were found in plasmids (Fig. 2g). These results suggest that the Rep_3-family genes were intra-cellularly transferred and expanded among plasmids so that replication initiator genes of all plasmids became Rep_3-family genes (Fig. 2g).

**Table 1 | Genome sequencing statistics**

| | *Persicobacter diffluens*[T] | *Persicobacter psychrovividus*[T] | *Aureibacter tunicatorum*[T] | *Fulvitalea axinellae*[T] | *Marivirga tractuosa*[T] | *Treponema saccharophilum*[T] | *Treponema bryantii*[T] |
|---|---|---|---|---|---|---|---|
| Total sequence length (bp) | 7,512,697 | 6,148,614 | 6,171,275 | 7,397,925 | 4,535,079 | 3,459,320 | 3,516,600 |
| Number of replicons or scaffolds | 16 | 13 | 9 | 15 | 2 | 4 | 2 |
| GC content (%) | 42.1 | 43.0 | 37.1 | 46.9 | 35.6 | 53.2 | 38.1 |
| Number of CDSs | 5354 | 4551 | 4910 | 5403 | 3732 | 2903 | 3006 |
| Coding ratio (%) | 80.3 | 84.0 | 87.4 | 85.1 | 88.4 | 88.2 | 92.7 |
| Number of rRNA genes | 9 | 6 | 6 | 6 | 6 | 3 | 12 |
| Number of tRNA genes | 165 | 173 | 141 | 126 | 39 | 60 | 41 |
| Replicon carrying rRNA genes | plasmid | plasmid | plasmid | plasmid | chromosome | plasmid | chromosome |

The "T" symbols show type strains.

## Rep_3-type plasmids significantly and specifically acquire *rrn* operons

Given that bacteria without chromosomal *rrn* operons always carried Rep_3-family genes on their *rrn* plasmids in our dataset, we hypothesized that Rep_3-family plasmids would have a propensity to carry rRNA genes. We examined the localization of *rrn* operons and Rep-family genes on plasmid contigs obtained from AnnoTree (GTDB r89)[30]. We found that *rrn* operons occur significantly more frequently on plasmids with Rep_3-family genes than on those with other *rep* genes ($p = 0.002$, Chi-square test) (Fig. 3a). Therefore, we argue that Rep_3-family genes have special characteristics that allow them to acquire and maintain *rrn* operons on a replicon.

We next hypothesized that Rep_3-family genes can allow their replicons to acquire not only *rrn* operons but also other essential genes. This was because, for example, the Rep_3-family genes could make plasmids extremely stable over generations by decreasing the plasmid loss rates, so that essential genes could have been maintained on plasmids. However, we found that Rep_3-family genes less frequently co-existed with universal single-copy genes in a plasmid than the other *rep* genes ($p < 0.001$, Fig. 3b).

## Rep_3-type plasmids stably maintained *rrn* operons due to their high-copy numbers

The fact that Rep_3-family genes can enable plasmids to specifically acquire and maintain *rrn* operons suggests that Rep_3-family genes may play a role in controlling plasmid copy numbers. This is because what characterizes rRNA genes compared to other essential protein-coding genes is their lack of translational-level amplification and high copy numbers. Because the *rrn* plasmid of *A. ureilytica* has a high copy number (10.9 plasmid copies per genome in stationary phase, Fig. 3c)[11], we analyzed short-read sequencing data to examine if the *rrn* plasmid of *T. saccharophilum*, which also recently lost chromosomal *rrn* operons, also has a high copy number. The copy number of the *rrn* plasmid of *Aureimonas* sp. AU20 were differently estimated from the previous study[11] (18.2 copies in the stationary phase). This difference could be attributed to discrepancies in culture techniques or methods to determine plasmid copy numbers. For example, the previous study used qPCR (*rrs/rpsB*), which can be affected by a PCR amplification bias. For comparison, we also estimated copy numbers of *rrn* plasmids with *rep* genes of bacteria with chromosomal *rrn* operons. We selected four strains whose short-read sequencing data were publicly available and that had small (<35 kb) contigs that encoded *rrn* operon(s) and Rep_3-family genes (*Bacillus* sp. OV322 and *Ureibacillus xyleni* JC22) or the other *rep* genes (*Eubacterium siraeum* DSM 15702 and *Tistlia consotensis* USBA 355) (Supplementary Fig. 4).

We estimated that Rep_3-type *rrn* plasmids of all the above species have high copy numbers (>10 copies, OV322 with *parA* gene was exceptionally ~8 copies) (Fig. 3c, Supplementary Fig. 4), whereas the

*rrn* plasmids with other *rep* genes had low copy numbers (<2 copies) (Supplementary Fig. 4). These results suggested that Rep_3-family genes can maintain plasmids at high copy numbers, even with the existence of *rrn* operons. High-copy number plasmids can be inherited stably even by stochastic segregations[31,32].

## Rep_3-type *rrn* plasmids of Persicobacteraceae decreased copy numbers but obtained partitioning mechanisms for stable inheritance

Next, we estimated copy numbers of the *rrn* plasmids of the four Persicobacteraceae species, which lost their chromosomal *rrn* operons much earlier. Interestingly, the copy numbers of their *rrn* plasmids were lower (four to eight copies) than those of species that recently lost their chromosomal *rrn* operons (Fig. 3c). Instead, *rrn* operons in the Persicobacteraceae species increased their copy numbers on the plasmids to two or three (Fig. 1a). In total, the copy numbers of *rrn* operons per cell remained constant (Fig. 3d). To our knowledge, no previous studies have observed that evolutionary pressure actively maintained copy numbers of plasmids and *rrn* operons[33].

Because we assumed that the high-copy numbers of the Rep_3-type *rrn* plasmids contributed to their stable inheritance, the copy number decrease of *rrn* plasmids in Persicobacteraceae may have needed to be compensated for by another mechanism for stable inheritance. Here, we found that *rrn* plasmids of *Persicobacter* spp. have *parAB* genes for active plasmid partitioning. On the other hand, the *rrn* plasmids of *A. tunicatorum*, *F. axinellae* and *T. saccharophilum* lack plasmid partition systems, as well as *A. ureilytica* and *O. saccharovorans*[11,18]. While *rrn* plasmids carrying *parAB* were synapomorphic in *Persicobacter*, *rrn* plasmids of *F. axinellae* and *A. tunicatorum* were outgroups in the 16S rRNA phylogenetic tree (Supplementary Fig. 5). Thus based on the Occam's razor principle, we estimate that the ancestor of Persicobacteraceae originally had *rrn* plasmids without *parAB* and acquired *parAB* in the ancestor of *Persicobacter*, which was likely the first empirical data that support a prediction that a plasmid obtains a partition system after the acquisition of essential genes[14].

## Losses of chromosomal *rrn* operons led to substantial increase of chromosomal tRNA genes

Next, we investigated genomic outcomes of losses of chromosomal *rrn* operons over short and long terms. Bacteria without chromosomal *rrn* operons are those that have the highest effective copy numbers of *rrn* operons per cell (Fig. 3d)[17,18], whereas related species of the four clades typically had only 1–5 *rrn* operons on their chromosomes (Fig. 1f, g and Supplementary Fig. 6).

We first focused on tRNA genes because rRNA gene numbers are known to correlate with those of tRNA genes as they constitute translation systems together[34,35]. As expected, tRNA gene numbers of

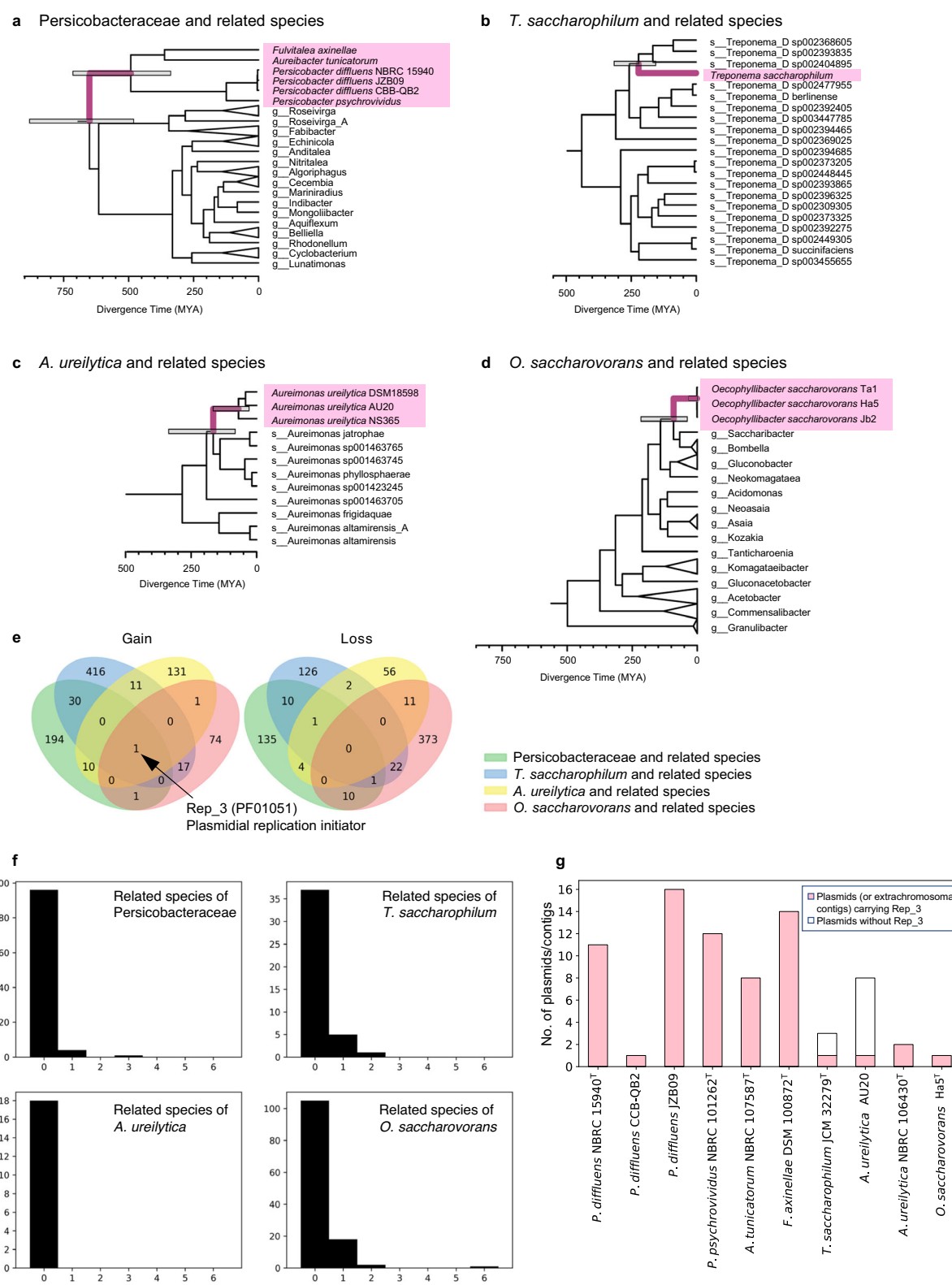

f

g

*A. ureilytica* and *T. saccharophilum* were significantly higher than those of related species (Fig. 4a and Table 1). The tRNA gene numbers of *O. saccharovorans* were not significantly greater than those of related species, but this may be because *O. saccharovorans* and its related species are undergoing substantial genome reduction (Supplementary Fig. 7)[36]. Notably, tRNA gene numbers of the Persicobacteraceae species were substantially high (128–173) (Fig. 4a and Table 1).

The substantial increase of tRNA gene numbers in the Persicobacteraceae species let us analyze which tRNA genes specifically increased in abundance. As mentioned above, tRNA$^{Ala}$ and tRNA$^{Ile}$ genes are resident on *rrn* plasmids and their copy numbers are estimated to be eight to twelve copies per cell. Copy numbers of chromosomal tRNA genes per amino acid ranged from three to fifteen and those per anticodon ranged from one to fifteen (Fig. 4b and

**Fig. 2 | Divergence times of bacteria without chromosomal *rrn* operons and events of those ancestors.** Genomic phylogenetic trees and estimated divergence times of **a** Persicobacteraceae species, **b** *T. saccharophilum*, **c** *A. ureilytica*, and **d** *O. saccharovorans*. Pink indicates bacteria without chromosomal *rrn* operons. Gray bars show confidence intervals of RelTime estimates which contain the actual time with 94% probability[90]. Purple branches are estimated timing of gain of *rrn* plasmids and loss of chromosomal *rrn* operons. Species and genus names follow GTDB taxonomy, but names of bacteria without chromosomal *rrn* operons follow IJSEM

for consistency with the main text. Phylogenetic trees including calibration points are shown in Supplementary Fig. 2. **e** Numbers of gene gain and loss events in ancestors that lost chromosomal *rrn* genes in the four clades. **f** Numbers of Rep_3-family genes in genomes of related species of bacteria without chromosomal *rrn* operons. **g** Numbers of plasmids (or extrachromosomal contigs) that carry Rep_3-family with or without *rrn* operons in bacteria without chromosomal *rrn* operons. Pink indicates plasmids (or extrachromosomal contigs) carrying Rep_3 in bacteria without chromosomal *rrn* operons.

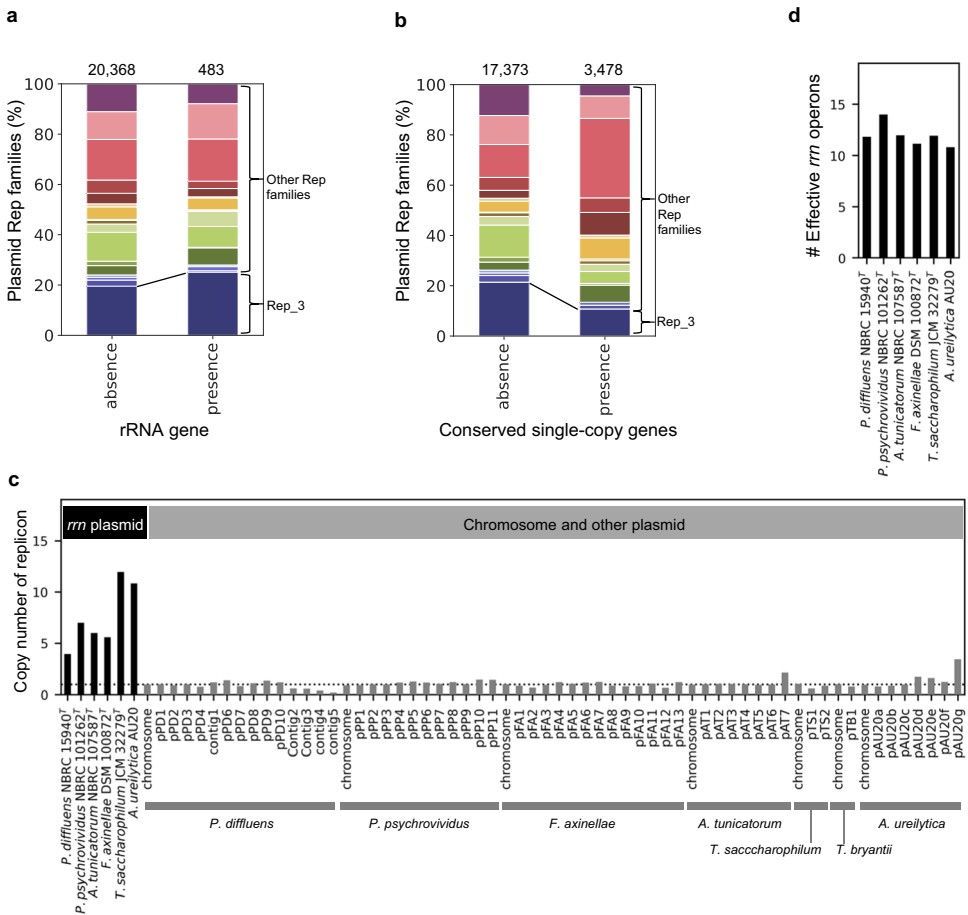

**Fig. 3 | Maintenance of *rrn* plasmids with Rep_3-family genes in bacteria without chromosomal *rrn* operons. a** Proportions of *rep*-family genes that are encoded on plasmids with (right) and without (left) rRNA genes. Numbers at the top of the panel show the total numbers. **b** Proportions of *rep*-family genes that are encoded on plasmids with (right) and without (left) conserved single-copy genes.

**c** Estimated relative copy numbers of the *rrn* plasmids and other replicons in bacteria without chromosomal *rrn* operons. The average values of the chromosome are one and shown as a dotted line. **d** Estimated effective copy numbers of *rrn* operons.

Supplementary Fig. 8a). These numbers are substantially different from those of related species with chromosomal *rrn* operons, one to two copies of tRNA genes per anticodon (Supplementary Fig. 8a). tRNA gene-copy number differences among amino acids (i.e., tRNA iso-acceptor families) likely reflect amino-acid compositions in the protein-coding sequences in genomes[37,38], because significant correlations were observed (Supplementary Figs. 8c, 9). Differences among anticodons (i.e., tRNA isodecoder families) were also attributed to codon biases[34] (Supplementary Fig. 8b). Minimal doubling time estimated using codon-usage bias[39] was also significantly shorter than that of related species (Fig. 4c). These results suggest that bacteria whose *rrn* operon is only on a plasmid were under selection pressure leading to faster growth rates and increased effective numbers of *rrn* operons and tRNA genes.

Finally, we investigated molecular mechanisms behind the increase in tRNA gene numbers in the Persicobacteraceae family. We

found that certain tRNA genes underwent tandem duplication on chromosomes, and each genome contains 20–25 tRNA gene clusters (Fig. 4d and Supplementary Fig. 8d). The largest tRNA gene cluster of *P. psychrovividus* (22 tRNA genes) is comparable in size to clusters belonging to a special group known as tRNA arrays[40–42]. A notable difference between the previously reported tRNA gene arrays and those of the Persicobacteraceae species was that the latter has a smaller repertoire of anticodons, possibly because the increase of tRNA gene-copy number in the Persicobacteraceae family is still ongoing (Supplementary Fig. 8d, e). Merging of different tRNA gene clusters also likely occurred in the Persicobacteraceae family, because the numbers of tRNA gene clusters were significantly lower than those of related species (Fig. 4d, Supplementary Fig. 8f). Because tRNA[Ile] genes (anticodon GAT) were found only within *rrn* operons on the plasmids of the Persicobacteraceae species, the increase in tRNA gene abundance on the chromosomes likely occurred after the transfer of

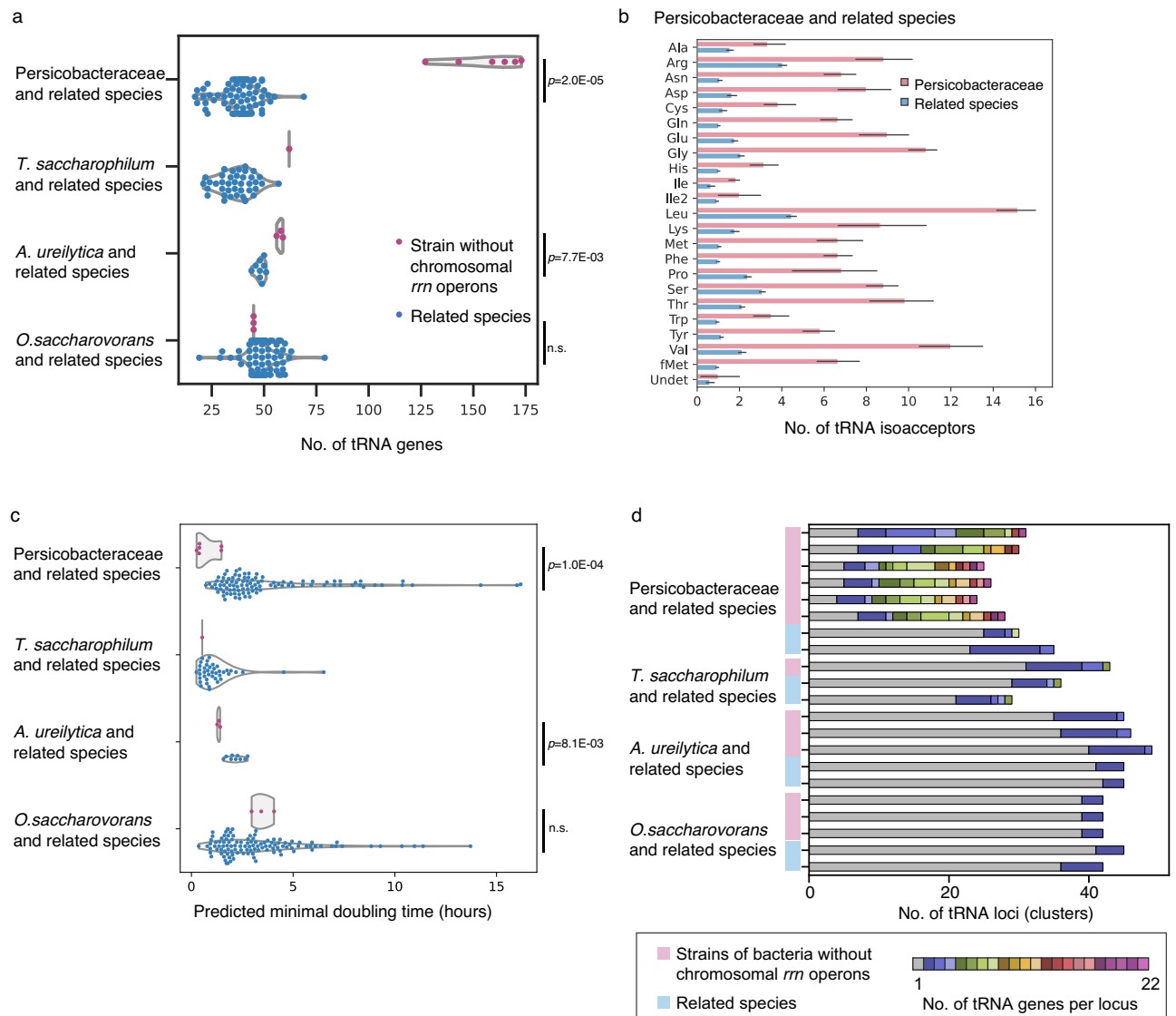

**Fig. 4 | tRNA gene-copy numbers in bacteria without chromosomal *rrn* operons.**
**a** Comparison with related species for the four independently evolved bacterial clades: Persicobacteraceae (*n* = 6) and related species (*n* = 101), *T. saccharophirum* (*n* = 1) and related species (*n* = 43), *A. ureilytica* (*n* = 3) and related species (*n* = 18), and *O. saccharovorans* (*n* = 3) and related species (*n* = 126). Each dot represents a genome. **b** tRNA gene numbers per isoacceptor family in the Persicobacteraceae (*n* = 6) and related species (*n* = 101). tRNA$^{Ile}$ (anticodon GAT) genes are exclusively on *rrn* plasmids, tRNA$^{Ala}$ genes are on both *rrn* plasmids and chromosomes, and the

others are exclusively on chromosomes. tRNA$^{Ile2}$ is an AUA codon-specific iso-leucine tRNA that has a CAU anticodon. Data are presented as mean values ±standard deviations. **c** Minimal doubling time estimated using gRodon[39]. Dataset of genomes is the same as in **a**. **d** Numbers of tRNA loci (gene clusters) per cluster size (tRNA gene numbers). **a**, **c** *P* values were calculated using Mann–Whitney *U* test (one sided). n.s. not significant (*p* > 0.05). Source data are provided as a Source Data file.

tRNA$^{Ile}$ genes to plasmids, unless tRNA$^{Ile}$ gene-copy numbers first increased on the chromosomes and subsequently decreased.

## Discussion

In this study, we found that chromosomal *rrn* operons were lost at least four times in the bacterial domain, where the Rep_3-family genes were revealed to be prerequisites. The long-term evolution of bacteria without chromosomal *rrn* operons has likely been supported by subsequent acquisition of plasmid partition systems. A notable evolutionary outcome was a substantial increase in tRNA gene numbers on chromosomes and decreased plasmid copy numbers (Fig. 5).

By assuming that the loss of chromosomal *rrn* operons occurred once in the common ancestor of each clade, we estimated that the Persicobacteraceae clade has survived throughout evolution without chromosomal *rrn* operons for >492 million years since the Paleozoic era. Even if we relax this criterion to allow three independent losses of

chromosomal *rrn* operons within the Persicobacteraceae clade, the loss is estimated to have occurred at 124–492 MYA (i.e., in the common ancestor of the genus *Persicobacter*) based on the shared genomic and plasmid structures in *Persicobacter* (Fig. 2a). We also argue that the unique plasmid and genomic characteristics of the four *Persicobacter* species strongly suggest that their *rrn* plasmids have evolved under the same evolutionary pressure as that on the *rrn* operons. This, in turn, supports the hypothesis under which the loss of their chromosomal *rrn* operons occurred in their common ancestor. Although there are debates on the accuracy of divergence time estimation in prokaryotes[43], our data prove that bacteria without chromosomal *rrn* operons can avoid extinction for an evolutionarily long term.

The convergent evolution of chromosomal *rrn*-operon losses strongly suggests that more clades lost chromosomal *rrn* operons in the bacteria domain. Above all, draft genomes did not allow us to list all bacteria without chromosomal *rrn* operons. Our bioinformatic search

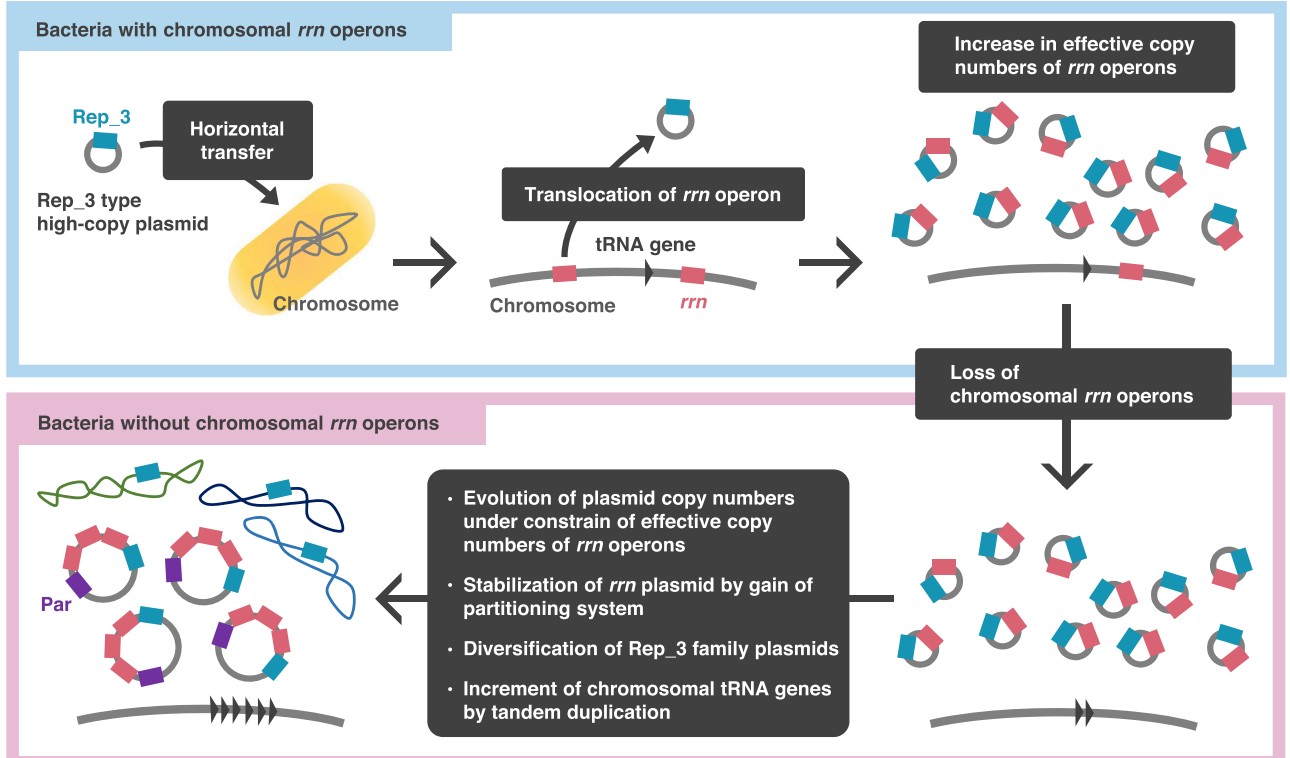

**Fig. 5 | Evolutionary model of bacteria without chromosomal *rrn* operons based on the findings of in this study.** (1) An ancestral bacterium obtains a prerequisite Rep_3-type plasmid by horizontal gene transfer. (2) An *rrn* operon is translocated to the Rep_3-type plasmid. (3) Effective copy numbers of *rrn* operons increase on the high copy number Rep_3-type plasmid. (4) Resultant evolutionary pressure leads to loss of chromosomal *rrn* operons, making the *rrn* plasmid indispensable to the bacteria. (5) Copy numbers of *rrn* plasmids decrease by keeping effective copy numbers of *rrn* operons, the *rrn* plasmid acquires a partitioning system for stable inheritance, and chromosomal tRNA genes undergo tandem duplications.

depended on the existence of *rep* genes, but there exist plasmids without *rep* genes[20]. In addition, our filtering criteria excluded plasmids that are large and/or have essential genes. Bioinformatic tools for detecting plasmid contigs may be used for detecting more cases[44]. Moreover, considering the fast increase in the number of publicly available high-quality metagenome-assembled genomes and single amplified genomes[45,46], it would soon be possible to search for vast numbers of uncultured bacteria for those without chromosomal *rrn* operons. For example, a recent long-read metagenomic study reconstructed a metagenome-assembled genome of *A. ureilytica* without chromosomal *rrn* operon[47]. In addition, a strain with a Rep_3-type high-copy *rrn* plasmid may lose chromosomal *rrn* operons in the future according to the present results.

The molecular mechanisms by which Rep_3-family genes maintain *rrn* plasmids at high copy numbers are of interest, and the molecular biology of Rep_3-family proteins merits exploration. We envision that, once their molecular mechanisms are clarified, Rep_3-family genes may be used as a synthetic biology tool for controlling plasmid functions. Plasmids with Rep_3-family genes are also known to encode iterons near the replication origins[48], which are direct repeats that control plasmid copy numbers by binding to Rep_3 family protein[49]. Thus, iterons and Rep_3 family protein might also play some role in maintaining the high copy numbers of *rrn* plasmids. While the order of events was inferred by the Occam's theorem (Fig. 5), the order of the Rep_3 stabilization and the *rrn*-operon acquisition is not clear. We argue that the *rrn*-operon acquisition may be the first because more Rep_3 hosts would have lost the chromosomal *rrn* operons if the Rep_3 stabilization was the first. Another limitation of this study would be that the observed plasmid copy numbers may be different from those in native conditions. Meanwhile, copy numbers of *rrn* operons, tRNA genes, and codon biases are indicators of growth rates, where the

number of tRNA genes correlates with the amino-acid composition of CDSs[34,37–39,50,51]. The speciality of the Persicobacteraceae clade may be due to differences in the time elapsed since the increase of the copy numbers of *rrn* operons; tRNA gene-copy numbers have been under strong evolutionary pressure to increase their copy numbers because of the increase of *rrn*-operon copy numbers.

The Persicobacteraceae species may also be of general interest in the context of plasmid biology, especially for those interested in plasmid evolution. Evolution of a specific plasmid can seldom be tracked for a very long time because of few conserved regions, high numbers of repetitive sequences, high rates of gene exchange, and many structural variants[52,53]. In the context of genome evolution, these species may also provide a unique platform to investigate the evolution of bacterial tRNA gene-copy numbers. During the long-term evolution of the Persicobacteraceae family, both partition system and biotin synthesis genes were transferred to their *rrn* plasmids (Fig. 1a). This may be because the Persicobacteraceae family needed to adopt *rrn* plasmids to produce more biotin, or because gene transfers between a chromosome and a stably inherited plasmid can occur by chance.

Finally, we may inquire why *rrn* operons have not returned to chromosomes again. One possible explanation is that other genomic characteristics (e.g., tRNA gene-copy number) that already adapted to the high-copy number plasmid *rrn* operons did not allow losses of plasmid *rrn* operons. It may also be possible that having *rrn* operons exclusively on plasmids provided evolutionary advantages, although the isolation sources of the four clades were diverse (marine organism or soil, Persicobacteraceae; rumen, *T. saccharophylum*; phyllosphere and air, *Aureimonas ureilytica*; and insect guts, *O. saccharovorans*). In addition to advantages previously proposed[11], one hypothesis would be that, because some antibiotics target rRNAs and rRNA genes on

different plasmid copies can harbor sequence diversity, high-copy number *rrn* plasmids may have been beneficial against toxin attacks[54]. Another hypothesis would be that, because transcription-replication clashes can act as a barrier to bacterial replication[55], *rrn* operons only on plasmids can make transcription and replication separated.

## Methods

### Genome dataset and search for genomes without chromosomal *rrn* operons

GBFF files of 86,822 bacterial genomes were downloaded from NCBI RefSeq on May 30, 2017. Among them, 406,111 sequences (contigs) encoded rRNA genes. Contigs coding Rep-family genes were selected by tBLASTn searches with 156 Rep sequences[20] at a threshold *e* value < 10⁻⁵, set to obtain hits for Rep genes on the *rrn* plasmid of *A. ureilytica*. Then, we selected genomes with a contig that encoded more than one full-length rRNA genes, encoded Rep-family genes, were <35 kb, and encoded no essential single-copy genes (bac120, GTDB-Tk v.0.3.2 identify[25]). The presence of full-length rRNA genes was based on RefSeq gene annotations. Genomes that had more than one contig that encoded rRNA genes were removed.

Accession numbers of *Persicobacter* spp. CCB-QB2 and JZB09 genomes were GCF_001274635.1 and GCF_001308105.1, respectively.

### Sample preparation, genome sequencing, assembly, and annotation

Supplementary Table 1 presents a summary of seven genomes sequenced in this study. These strains were provided by the Japan Collection of Microorganisms (JCM), NITE Biological Resource Center (NBRC), and Deutsche Sammlung von Mikroorganismen und Zellkulturen GmbH (DSMZ). Strains of *T. saccharophilum, T. bryantii*, and Persicobacteraceae and related species were incubated using *Treponema saccharophilum* medium (DSMZ 323), *Treponema bryantii* medium (DSMZ 159), and Marine Broth medium (Difco, Tokyo, Japan), respectively. Cells in stationary phase were collected and total DNA was extracted using DNeasy PowerSoil Kit (QIAGEN) in accordance with the manufacturer's instructions.

Genomes were sequenced using PacBio RSII (Pacific Biosciences) and HiSeq X ten (2 × 150 bp). PacBio RSII libraries were size-selected by BluePippin (6–50 kb for *A. tunicatorum, P. diffluens, T. saccharophilum*, and *T. bryantii*, and 17–50 kb for *F. axinellae* and *P. psychrovividus*). Primer sequences of HiSeq reads were trimmed using Trimmomatic v.0.33[56]. Hybrid assembly was conducted using Unicycler v.0.4.8-beta (normal mode)[57], and/or HGAP v.2.2.0[58] provided in DDBJ annotation Pipeline with Pilon v.1.23[59]. Finishing was performed using GenomeMatcher[60], minimap2 v.2.0-r191-dirty[61], and Bandage v.0.8.1[62]. Completeness of the genomes was assessed using BUSCO v.4.0.2[63] on the Cytophagales_odb10, Spirochaetales_odb10, Rhizobiales_odb10, and Rhodospirillales_odb10 orthologue datasets. Although the assembly status of *Persicobacter diffluens* was scaffold, its completeness was comparable with complete genomes of other Persicobacteraceae species. Circular contigs were labeled as chromosomes or plasmids (Supplementary Table 1).

Genome annotation was performed using DFAST v.1.1.0[64] and gene prediction was using Prodigal v.2.6.3[65]. Barrnap v.0.8 (https://github.com/tseemann/barrnap) and tRNAscan-SE v.2.0.6[66] were used for rRNA and tRNA gene prediction. KofamScan v.1.1.0 (with KOfam v.2020-01-06[67]) and InterProScan v.5.24-63[68] (with Pfam v.31.0[27] and TIGRFAMs v.15.0[69]) were used for functional annotation of CDSs. For *Persicobacter* spp., *rep* genes were numbered according to sequence similarity and synteny as shown in Fig. 1e.

### Phylogenetic analysis

In the phylogenetic analyses based on concatenated rRNA genes (*rrs, rrl, rrf*) and single-copy genes of the Persicobacteraceae species and *T. saccharophilum* and their related species, 17 and 19 genomes were used, respectively (Supplementary Data 1). Nucleotide sequences of the rRNA genes were aligned using MAFFT v.7.273[70] and curated using TrimAl v.1.2rev59[71] (strict mode for *Treponema*, and gappyout mode for Cytophagales). Partitioned analysis of the rRNA genes with the best substitution models for each alignment was conducted using IQ-TREE v2.0.3 and ModelFinder[72–74]. The alignments of 4467 (1512, 2848, 107) or 4558 (1529, 2919, 110) nucleotides, respectively, were subjected to phylogenetic tree reconstruction using IQ-TREE v2.0.3 with the best substitution model and 1000 bootstrap replicates. The similarity between the topologies of core genes and rRNA genes was tested statistically by likelihood tests with IQ-TREE[75–79].

Phylogenetic analysis based on amino-acid sequences of single-copy core genes was performed using bcgtree v.1.0.10, which uses hidden Markov models and performs a partitioned maximum-likelihood analysis[80], and RAxML v.8.2.9[81] with bootstrap trial set to 1000. In total, 85 or 88 CDSs were used, respectively.

For phylogenetic analysis of Rep genes including Rep_3-family genes (PF01051), we used BLASTp (ncbi-blast-2.10.0+, *e* value < 10⁻⁵, query coverage >0.5, identity >30%) against nr database by querying 93 amino-acid sequences of Rep_3-family genes from genomes without chromosomal *rrn* operons, and obtained bacterial 2,723 sequences. The representative sequences of 1413 clusters generated using CD-HIT v.4.8.1 (95% identity)[82] (Supplementary Data 5) were aligned using MAFFT and curated using TrimAl (gappyout) and pgelimdupseq v.2.0.2016.09.06 (208 positions). The alignment was subjected to phylogenetic tree reconstruction using FastTree v.2.1.10[83].

Phylogenetic tree visualization was performed with iTOL v.6[84], and NCBI taxonomy was assigned using ETE 3[85].

### Divergence time estimation

We obtained g_Aureimonas and g_Treponema_D phylogenetic trees from GTDB v.89.0[24]. Phylogenetic trees of the Persicobacteraceae species and *O. saccharovorans* were reconstructed using GTDB-Tk v.0.3.2 gtdbtk de_novo_wf[25]. These phylogenetic trees consisted of bacteria without chromosomal *rrn* operons, related species, calibration species for divergence time estimation, outgroups, and those for stability of phylogenetic trees shown in Supplementary Data 6. The related species were f_Cyclobacteriaceae for Persicobacteraceae, g_Treponema_D for *T. saccharophilum*, g_Aureimonas for *A. ureilytica*, and f_Acetobacteraceae for *O. saccharovorans*. Divergence time was estimated using RelTime-Branch Lengths[26,86] for the GTDB or GTDB-Tk phylogenetic trees. In RelTime, branch-specific relative rates are estimated by applying equal elapsed time periods of separation of two sister lineages from their most recent common ancestor[26] and were used for estimating branching time of bacteria[87]. Calibration constraints were adopted from a previous study[88].

### Ancestral gene-content reconstruction

We obtained subtrees consisting of bacteria with chromosomal *rrn* operons and related species from the phylogenetic trees used for divergence time estimation and constructed gene-family tables using Pfam[27]. PastML v.1.9.24[28] was used to predict gains/losses of gene families using presence/absence tables. CAFE v.4.2.1 was used to predict expansion and contraction of gene numbers of gene families that were predicted to be present at the root[29]. Analyses using KEGG[67] and TIGRFAMs[69] were also performed.

### Dataset of plasmids encoding Rep-family genes

The 156 Rep proteins[20] were contained in 32 Pfam families (*e* value < 10⁻⁵, Supplementary Data 7). Among them, 17 families have functions clearly related to plasmid replication and are in AnnoTree[30] with GTDB r89.0. We obtained 21,716 contigs that had those Rep-family genes from 9728 genomes. We removed chromosomes that had Rep-family genes by filtering genomes containing single contigs, and the largest contigs in complete genomes. We also calculated numbers

of universal single-copy genes by the identify command of GTDB-Tk and removed contigs >one unique gene for contigs whose lengths were <54 kb, more than four unique genes for contigs whose lengths were 54 kb or more and <800 kb, and more than nine unique genes for contigs whose lengths were 800 kb or more. The annotation of rRNA genes was performed using Barrnap v.0.8.

## Estimation of plasmid copy numbers

Plasmid copy numbers were calculated by mapping short reads to genomes and dividing average coverage depths of plasmids by those of chromosomes. For *T. saccharophilum* and the Persicobacteraceae species, HiSeq reads used for genome assembly were retrieved. For *Aureimonas* sp. AU20, we conducted sequencing using HiSeq X Ten (2 × 150 bp). We obtained genomes and short-read data for *Bacillus* sp. OV322 (GCF_900112495.1 and SRR4235142), *Ureibacillus xyleni* JC22 (GCF_900217795.1 and SRR6007419), *Eubacterium siraeu* DSM 15702 (GCF_000154325.1 and SRR15171209.1), and *Tistlia consotensis* USBA 355 (GCF_900177295.1 and SRR5194480) from SRA and NCBI RefSeq. These strains were selected from our in-house dataset entries that contained *rrn* operons in the chromosome and a small plasmid (<35 kb). Whether the contigs were chromosomal or not was assessed by Bandage[62]. The threshold for distinguishing chromosomal and plasmid contigs was the same as above. Coverage depths were calculated using Bowtie2 v.2.2.7[89] and bbmap v.38.34 (https://www.osti.gov/biblio/1241166).

## tRNA clusters and codon bias *S*

A group of tRNA genes was considered a cluster if the tRNA genes were adjacent to each other at ≤500 bp. Spearman's correlation coefficient between the numbers of tRNA genes on chromosomes and the amino-acid compositions of all CDSs were calculated. Codon bias $S$[34] was calculated using a python script by referring to R-package v.2.24.0 *sscu*. The growth-rate potential was estimated using gRodon R-package v. 2.3.0[39]. Forty highly expressed genes were selected according to a previous study[34] and used for calculating codon bias $S$ and growth-rate potential. Statistical significance was assessed using Mann-Whitney $U$-tests.

## Reporting summary

Further information on research design is available in the Nature Portfolio Reporting Summary linked to this article.

# Data availability

The sequences reported in this paper have been deposited in the DNA Data Bank of Japan (DDBJ) database, http://www.ddbj.nig.ac.jp (accession numbers are listed in Supplementary Table 1). Other data generated in this study are provided in the Supplementary Data and Source Data files. Source data are provided in this paper.

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

## Acknowledgements
We thank Motomu Matsui, Seishiro Aoki, Joel Nitta, and Ken Kuroki for their helpful comments, and Naomi Sakurai for helping in culturing the anaerobic bacteria. This study was supported by JSPS KAKENHI Grant Numbers 18J00444 (to M.A.), 16H06279 (to W.I. and A.T.), 19H05688, and 22H04925 (to W.I.) and JST Grant Number JPMJCR19S2 (to W.I.). M.A. was supported by JSPS Research Fellowships.

## Author contributions
M.A. and W.I. designed the research; M.A. did the genome finishing; M.A. performed experiments; M.S., M.O., M.T. cultured anaerobic bacteria; A.T. performed genome sequencing; M.A., S.Y. and S.C. analyzed data; W.I. supervised the study; and M.A. and W.I. wrote the paper.

## Competing interests
The authors declare no competing interests.
