## [Peer Review File · Nature Communications]

Bacteria can maintain rRNA operons solely on plasmids for
hundreds of millions of yearsReviewer #1 (Remarks to the Author):

In this manuscript, Anda and colleagues expand on previous work to investigate the presence of rRNA operons (*rrn*) solely on plasmids. The subject is a compelling one, first because *rrn* have been taken as *the* identifying feature of bacteria through 16S sequencing, so their absence from the chromosome and presence on plasmids implies that *rrn* can be less tightly linked with the chromosome than assumed; and second, because plasmids can affect gene evolution in various ways, the prospect of such mechanisms governing *rrn* evolution is an intriguing one. The manuscript, including figures, was well presented, however there are a few areas that would benefit from further work.

My main issue: The key finding is that the *Persicobacteraceae* lack chromosomal *rrn*, and the last common ancestor of this clade existed >492 mya. The authors propose a more conservative estimate (lines 305ff) from the possibility that even if it is just the common ancestor of *Persicobacteria* that lacks chromosomal *rrn*, *rrn* has been exclusively plasmid-borne for >124 my. But as the *Persicobacteraceae* are relatively undersampled in databases (I understand that the authors are analysing all available *Persicobacteraceae*), it seems possible that more intense sampling would reveal members of this clade with chromosomal *rrn*, which would bring forward the time for which we can infer organisms can persist without chromosomal *rrn*. Similar issues apply to the other genera studied (line 164ff), but are exacerbated as there are even fewer divergent non-chromosomal *rrn* members in these groups. The conclusions, particularly in the abstract, ought to be toned down.

Further major issues/suggestions:

- There is a large body of work looking at the evolutionary and ecological drivers of *rrn* copy number (e.g. doi: 10.1038/nmicrobiol.2016.160). Carriage of the *rrn* operon by plasmids seems an elegant evolutionary solution to the problem of adaptively modulating *rrn* copy number. The authors should consider discussing their results in the context of *rrn* copy number evolution/ecology.
- The authors propose that by guarding against ribosome-targeting antibiotics, the heterogeneity in *rrn* operons made possible by plasmids helps protect bacteria against interference competition (line 345ff). While beyond the scope of the paper, it would be interesting to model this hypothesis, particularly in the context of segregational drift (doi: 10.1093/molbev/msy225). I have another hypothesis. Ribosomes are transcribed at high rates (since, as the authors point out, there are not opportunities for translational amplification). Transcription-replication clashes can act as a barrier to bacterial replication (see e.g. 10.1146/annurev-micro-090817-062514). By harbouring *rrn* on multiple plasmids, transcription and replication could be separated. The authors might like to consider discussing this possibility.
- Line 182. Can you infer the order of events? Is there any evidence that the plasmid was stabilised in the lineage before acquiring *rrn*? There is some experimental data suggesting that essential genes are more likely to be acquired by the plasmid before it becomes stable (i.e. ref 10, doi: 10.1371/journal.pgen.1009656) — contrasts and implications could be discussed further in the discussion. Similarly, how was order of *parAB* gene acquisition inferred in line 244ff?
- Line 191. How does 2g show this? Fig. 2g is also not clear: in the legend, doesn't the pink refer to the contigs (not bacteria)?
- Line 196. How do you show they 'strictly require'? Isn't it just correlation?
- Fig. 3a and b. What database was used?
- Line 686. Define 'conserved single-copy genes'.
- Figure 4c. Not clear what interpretation of this is — this figure could be better explained in the text.
- Is it possible to infer when the plasmid *rrn* genes were gained, and add this information to the tree in Figures 1 or 2?
- The discussion of the tRNA results is lacking. For example, why do the authors think that some tRNAs expand on the chromosome, whereas others are on the plasmids?
- Line 355. E-value of 1e-5. What identity and length threshold is this (approximately) and how sensitive are findings to varying this parameter?
- Line 698. 94% is a peculiar threshold. What is the rationale?
- Table 1. Are these assemblies complete (use of 'scaffold' suggests they may be draft genomes)?
- Ext. dat. Fig. 2. has non-prokaryotes on the tree (genus *Drosophila*, *Athalia*), presumably

because they were the source for candidate species, but this should be explained.

Minor/stylistic/typographical suggestions:

- Line 52. Vague — are you referring to the plasmid genomes, or the bacterial genome as a whole?
- Line 61. "Potential of encoding" unclear. Suggested rewrite: "by the fact that chromosomes encode essential genes".
- Line 62. How to determine the 'most' essential? Are these more important than e.g. DNA polymerase? Reference required.
- Line 71. Is reference 13 relevant here?
- Line 103. How did you *confirm* that they didn't make false positives or false negatives?
- Paragraphs starting line 111ff. Make it clear here that these paragraphs are referring to the Genbank genome references.
- Line 201. Or they have more opportunity to acquire such genes?
- Line 223 onwards. Copy number can be variable though — these counts are from cultures grown specifically for DNA extraction, which might favour different numbers from those in the natural habitat, which should be acknowledged.
- Fig 1e. Does black indicate no match? White would be more intuitive.
- Fig 3c. A dotted line for CN=1 would help orientate readers.
- Fig. 4. Are the data presented here corrected for plasmid copy number?
- Line 247. Not clear. Do *A. ureilytica* and *O. saccharovorans* have plasmid partition systems or not?
- Data set — typographical error ('withough' should be 'without').

Reviewer #2 (Remarks to the Author):

In this manuscript, Anda and colleagues systematically survey the NCBI RefSeq database to uncover novel bacterial clades that carry their *rrn* operons exclusively on plasmids. This is interesting because it challenges the assumption that essential genes should be carried preferentially on the chromosome to ensure a stable inheritance. First, the authors uncover four new bacterial clades that add to the two previously known clades (*Aureimonas* spp and *Oecophyllibacter* spp, refs 11 & 18). Then they demonstrate that *rrn*-plasmid associations are stable over evolutionary timescales. They also show how *rrn*-Rep_3 plasmids and their host chromosomes have co-evolved, particularly by increasing the number of available tRNA genes.

I am not a bioinformatics expert, so my review focuses on the biological significance of the results, particularly regarding plasmid biology. The results are potentially interesting, but I had trouble understanding some parts of the manuscript, and some decisions regarding data analyses seem arbitrary. Perhaps a more detailed methods section would help in justifying/explaining the author's choices (see below for some specific examples). Some sentences are also hard to understand because of language issues.

Specific comments

- The criteria for finding *rrn*-containing plasmids warrants a more detailed explanation. For instance, why <35 kb? There are plasmids certainly larger than that. Furthermore, after reading the methods, it is unclear how the presence of 'full-length rRNA genes' is determined and why contigs containing single-copy essential genes were discarded (also, what defines a single-copy essential gene?)
- I wonder how using specialized software to identify plasmid sequences, such as plasmidfinder (PMID: 31584170) or mob-suite (mob-typer; PMID: 30052170), would affect the author's results.
- In certain bacterial species, it is not uncommon to find plasmids with multiple replicons. It would be interesting to know if the Rep_3 *rrn* plasmids carry a single or multiple replicons

- The discussion section is vague, with some paragraphs being extremely speculative. It may be better to include a thorough discussion of the limitations of the work.
- I need help understanding the last sentence of the abstract (L54). Please re-write.
- Please revise the taxonomic nomenclature according to recent changes (e.g. Proteobacteria = Pseudomonadota)
- L145. Since *rrn* operons exclusively located on plasmids were already discovered in two species (refs 11 and 18), this sentence seems an overstatement. Tone down or remove.
- Supp. Table. 'withough'
- Figure 2 legend: 94% confidence? It is more common to plot 95% IC. Is there a reason to choose 94 instead?
- The term steal is repeatedly used throughout the article. However, a less anthropomorphic term would be more appropriate. Transfer?
- L187: How a related species is defined? Figure 2f: I am unsure that the term 'copy number' is appropriate here.
- L191: I do not see how the authors reached that conclusion. From what I see, several strains of the *Persicobacteraceae* carry only Rep-3 plasmids, but that does not mean that 'Rep_3-family genes were intracellularly transferred and expanded among plasmids' The authors should better explain this claim
- L222: Please refer here to Extended data figure 3
- L344: lumen?

Reviewer #3 (Remarks to the Author):

The authors were initially inspired by reports of sole *rrn*-operons on plasmids in *A. ureilytica* and *O. saccharovorans*, and subsequently investigated the presence of this phenomenon in *Persicobacteraceae*. They identified six strains from four genera that harbored plasmid-borne *rrn*-operons, potentially as a result of a translocation event from the chromosome. The plasmid-borne *rrn*-operons were found to have a higher copy number compared to those on the chromosome, which corresponded to an increase in the number of tRNA gene clusters on the chromosomes.

1. They found three additional genomes has sole *rrn* operons in plasmids out of 86,822 genomes (0.003%). This result means such case is really rare, at least based on the dataset they used. *Persicobacter* spp (related *Persicobacter diffluens*); *Treponema saccharophilum*
2. Based on the bioinformatic analysis, they further explored *P. diffluens* relatives: *P. psychrovividus*, *Aureibacter tunicatorum*, and *Fulvitalea axinellae*. In total they found six strains in such case and confirmed two strains (*P. diffluens* NBRC 15940T and 128 *T. saccharophilum* JCM 32279T) by sequencing. Turn out *Persicobacteraceae* all in such case.

Taken together, this manuscript reports an interesting 'exception to the rule' on the evolution of plasmid gene content.

Comments according to their order in the manuscript:

Lines 66, 167. The citation of refs 9,10 as contradicting the findings of *rrn* on plasmids is not entirely justified. I suggest that the authors read those publications carefully. In essence, these studied explain why essential genes in plasmids should be rare, which is exactly what this study

shows – *rrn* genes were found in 0.003% of all analyzed genomes. I would argue that this study is rather in agreement with both refs 9 & 10 (but for slightly different reasons since ref 10 at least deals with protein coding genes and dose effect, which is less relevant for RNA genes). Line 165-167: “It does challenge the belief that essential gene cannot be maintained stably on plasmids.” The current state is, a second copy of essential genes cannot be maintained stably on plasmids, sole essential gene can be maintained on plasmids. Again, please read carefully refs 9,10.

Line 102: They focus only on small plasmids containing *rrn* operons. (Contig size <35 kb) How does this cut-off value come up?

Line 675 (Figure 1): I checked the genome of *Persicobacter* sp. JZB09, the authors reported here the plasmid of size 30,222 bp is actually the smallest plasmid out of 16 plasmids in the same strain, this plasmid is also named JZB09-Plasmid16. In RefSeq the accession number of this contig is NZ_CP012859.1, which has been removed later by NCBI stuff. From our experience such ‘removal’ of plasmids may occur occasionally and these correspond (most likely) to tackling of in assembly error. In other words, my interpretation would be that this contig was erroneously annotated as a plasmid to begin with and is actually part of the chromosome. Please confirm the state of this assembly.

Line 675 (Figure 1): I noticed the synteny of *rrn* operons in Figure 1e, would it be possible using more rRNA genes to build the tree in Figure 1f instead of only 16S rRNA gene? Common structure: *rrs* (16S rRNA)-*trnI* (tRNA^{Ile})-*trnA* (tRNA^{Ala})-*rrl* (23S rRNA)-*rrf* (5S rRNA)

Line 149-156 The similarity between the topologies of core genes and 16s rRNA should be tested statistically. This can be done, e.g., with IQtree (using the core genes topology as a user tree for the 16 rRNA alignments and performing a likelihood test).

Line 148: Admittedly, I am not a fan of timed phylogenies for ancient splits in prokaryotes. I prefer not to comment extensively on this part. I think that it is sufficient to state that, e.g., the plasmid acquisition occurred before/after a specific species divergence event. I also note that the dating error bars in Fig. 2a-d span many ancestral nodes so I would not base any conclusion based on that inference alone. I could only recommend to read the publications of those dating methods carefully in order to better understand their assumptions and caveats.

Line 169: “stole”? I cannot imagine how did these small plasmids stole *rrn* operon from chromosome, there is no protein related to any MGEs on the same plasmid; through recombination? I recommend using terminology from the field of molecular evolution here – that is – either the operon was translocated from the chromosome to the plasmid, or an alternative copy was acquired (as suggested in Line 187 if I get it right) and subsequently the chromosomal operon was lost.

Line 187 & line 754: Extended Data Fig. 2. I don’t understand how were the transfer events inferred? marked with stars. And Plasmid pJZB16 is a *rrn*-plasmid, isn’t it? Why it has been marked as the other plasmid with an empty triangle? Could the authors supply the sequences, alignment and tree file for the phylogenetic trees in extended data Fig. 2?

Line 188: ‘Holders’ -> better use ‘Hosts’

Line 191-193: How did a replication initiation protein transfer between plasmid intracellularly? Based on the phylogenetic tree in extended Data Fig. 2?

Line 196-198: “we hypothesized that Rep₃-family plasmids would have a propensity to carry rRNA genes.” According to the phylogenetic tree in extended Data Fig. 2, I don’t see any clue for this hypothesis. Also – Fig. 3a suggests that the authors performed a test to compare the presence of rRNA gene among all Rep types (i.e., a 2xn contingency table) while the more suitable analysis to test this hypothesis would be with a 2x2 design – i.e., Rep₃-family versus all the rest.

Line 195-208: This part is very controversial, what is the special character of Rep_3-family? I tend to believe it's not only due to the plasmid but also the host. It's a neutral event, not a destiny that Rep_3-family plasmids will acquire any essential genes from chromosomes for sure. The following test for presence of core gene is not clear to me.

Line 210 - 230: Estimating the relative plasmid copy number from short read sequencing is legit (and it would be good to write here something about the comparison with ref. 11). The assessment of chromosome copy number (or ploidy) is very problematic here – I don't see anything in the methods that can explain how these number were obtained. The part about the plasmid high copy number being advantageous for having the *rrn* on a plasmid should be in the discussion.

Line 212: "Rep_3-family genes may play a role in controlling plasmid copy numbers" – Better place in the discussion part.

Line 227-228: "High-copy number plasmids can be inherited stably even by stochastic segregations." – this assumption (the authors did not test the stochastic segregation/passive partition) aligns with the common view in the field and would be still good to have here a citation.

Line 237: "In total, the copy numbers of *rrn* operons per cell remained constant (Fig. 3d)" – Fig.3d shows something else which I'm not sure how is related to this sentence.

Line 238-239: "To our knowledge, no previous studies have directly observed that evolutionary pressure actively maintained copy numbers of plasmids and *rrn* operons." – I am not sure that the authors show here a direct observation for that statement either. That is – they do not test the effect of evolutionary pressure on the plasmid copy number.

Line 244-245: "which was likely the first empirical confirmation of a prediction that a plasmid obtains a partition system after acquisition of essential genes" – I don't think that the authors showed empirically that the acquisition of the *parAB* occurred after the *rrn* acquisition. This may be 'inference' from phylogenetic trees (and the such) and should be communicated as such.

Line 248-252: better move this to the discussion.

Line 268-270: again – better to move to the discussion. This is not a clear result and is somewhat speculative.

Line 254: The last section on tRNA is also not very convincing in the absence of a 'contrast' to compare those observations to a group without plasmid-encoded *rrn*. This maybe as well be a *Persicobacteraceae* specific phenomena. The history and the mechanism were not clear to me.

Lines 299-304 and Fig. 5 – Im not sure that I have seen clear evidence for a traslocation of the chromosomal *rrn* to the plasmid rather than an acquisition of a 'foreign' *rrn* and loss of the chromosomal copy.

Line 313-314 – considering the parent of gene presence/absence, I would argue that the loss of the chromosomal copy is the most parsimonious event, without invoking arguments on selection pressure in the past (which are unknown).

The discussion overall includes some almost 'off topic' subjects like Lines 315-322. Including the topics discussed in the results section will make it more interesting and to the point.

The last discussion paragraph is quite speculative. I would say that having *rrn* only on a plasmid is a 'super addiction mechanism' that requires no further explanations for the plasmid stability in the host.

Reviewer #1 (Remarks to the Author):

In this manuscript, Anda and colleagues expand on previous work to investigate the presence of rRNA operons (*rrn*) solely on plasmids. The subject is a compelling one, first because *rrn* have been taken as ***the*** identifying feature of bacteria through 16S sequencing, so their absence from the chromosome and presence on plasmids implies that *rrn* can be less tightly linked with the chromosome than assumed; and second, because plasmids can affect gene evolution in various ways, the prospect of such mechanisms governing *rrn* evolution is an intriguing one. The manuscript, including figures, was well presented, however there are a few areas that would benefit from further work.

My main issue: The key finding is that the *Persicobacteraceae* lack chromosomal *rrn*, and the last common ancestor of this clade existed >492 mya. The authors propose a more conservative estimate (lines 305ff) from the possibility that even if it is just the common ancestor of *Persicobacteria* that lacks chromosomal *rrn*, *rrn* has been exclusively plasmid-borne for >124 my. But as the *Persicobacteraceae* are relatively undersampled in databases (I understand that the authors are analysing all available *Persicobacteraceae*), it seems possible that more intense sampling would reveal members of this clade with chromosomal *rrn*, which would bring forward the time for which we can infer organisms can persist without chromosomal *rrn*. Similar issues apply to the other genera studied (line 164ff), but are exacerbated as there are even fewer divergent non-chromosomal *rrn* members in these groups. The conclusions, particularly in the abstract, ought to be toned down.

Response (R1C1): Thank you for your overall positive evaluation and constructive comments on our manuscript. While the shared genomic and plasmid structures within the *Persicobactor* genus will support the common origin assumption even if more genomes will become available, we agree that more intense sampling is

demanded to decipher their evolutionary histories in more details. At this revision, we have revised the abstract (LINE 45) and discussion (LINES 317-318, 322-324) to make these points clearer.

Further major issues/suggestions:

There is a large body of work looking at the evolutionary and ecological drivers of *rrn* copy number (e.g. doi: 10.1038/nmicrobiol.2016.160). Carriage of the *rrn* operon by plasmids seems an elegant evolutionary solution to the problem of adaptively modulating *rrn* copy number. The authors should consider discussing their results in the context of *rrn* copy number evolution/ecology.

The authors propose that by guarding against ribosome-targeting antibiotics, the heterogeneity in *rrn* operons made possible by plasmids helps protect bacteria against interference competition (line 345ff). While beyond the scope of the paper, it would be interesting to model this hypothesis, particularly in the context of segregational drift (doi: 10.1093/molbev/msy225). I have another hypothesis. Ribosomes are transcribed at high rates (since, as the authors point out, there are not opportunities for translational amplification). Transcription-replication clashes can act as a barrier to bacterial replication (see e.g. 10.1146/annurev-micro-090817-062514). By harbouring *rrn* on multiple plasmids, transcription and replication could be separated. The authors might like to consider discussing this possibility.

Response (R1C2): Thank you for this insightful comment. We have added discussion on the possible role of *rrn* copy number evolution (LINES 350-353) and on the possible advantage in transcription-replication clashes (LINES 375-377).

Line 182. Can you infer the order of events? Is there any evidence that the plasmid was stabilised in the lineage before acquiring *rrn*? There is some experimental data suggesting that essential genes are more likely to be acquired by the plasmid

before it becomes stable (i.e. ref 10, doi: 10.1371/journal.pgen.1009656)
— contrasts and implications could be discussed further in the discussion.
Similarly, how was order of *parAB* gene acquisition inferred in line 244ff?

Response (R1C3): The acquirement of a Rep_3 plasmid was estimated to be a prerequisite based on the Occam's razor principle, but we have no evidence on when the plasmid was stabilized. Thus, according to your comment, we have added a discussion on this issue (LINES 345-348). Regarding *parAB*, *rmn* plasmids carrying *parAB* are synapomorphic in *Persicobacter*, where *rmn* plasmids of *F. axinellae* and *A. tunicatorum* are the outgroups in the 16S phylogenetic tree (Extended Data Fig. 5). Thus, it is estimated that the ancestor of the Persicobacteraceae originally had *rmn* plasmids without *parAB* and acquired *parAB* in the ancestor of *Persicobacter*. We have clarified this in the text (LINES 254-258).

Line 191. How does 2g show this? Fig. 2g is also not clear: in the legend, doesn't the pink refer to the contigs (not bacteria)?

Response (R1C4): We thank the reviewer for carefully and kindly pointing out these issues. As Fig. 2g alone was not sufficient to show that, we have added Extended Data Fig. 3 and a relevant explanation in the main text (LINES 190-193). We also apologize for the error in the legend of Fig. 2g. The error has now been corrected.

Line 196. How do you show they 'strictly require'? Isn't it just correlation?

Response (R1C5): We have accordingly changed the relevant text (LINES 199-200).

Fig. 3a and b. What database was used?

Response (R1C6): We used datasets of plasmids selected from GTDB r89.0 as described in 'Dataset of plasmids encoding Rep-family genes' in Materials & Methods (LINES 469-479).

Line 686. Define 'conserved single-copy genes'.

Response (R1C7): We have clarified the definition in the legend of Fig. 1. From the 107 essential single-copy core genes, 85 or 88 conserved CDSs were extracted by bcgtree v.1.0.10 (Supplementary Information 2).

Figure 4c. Not clear what interpretation of this is — this figure could be better explained in the text.

Response (R1C8): We have accordingly added an explanation and Extended Data Fig. 9 (LINES 350-352). We have moved the previous Figure 4c to Extended Data Fig. 8c.

Is it possible to infer when the plasmid *rrn* genes were gained, and add this information to the tree in Figures 1 or 2?

Response (R1C9): The timing of gain of plasmid *rrn* was shown as purple branches in Fig. 2a-d, but this explanation was erroneously missing. We thank for this comment and have revised the legend of Fig. 2a-d.

The discussion of the tRNA results is lacking. For example, why do the authors think that some tRNAs expand on the chromosome, whereas others are on the plasmids?

Response (R1C10): We have added a discussion on tRNA genes (LINES 350-355). In Persicobacteraceae, most tRNA genes are located on chromosomes (Extended Data Table 1), and tRNA-Ile (GAT) and tRNA-Ala (TGC) are found only in the *rrn* plasmid (the only exception is tRNA-Ala (TGC) in *Aureibacter tunicatorum*, which is present in the *rrn* plasmid and the chromosome (Fig. 1a & Extended Data Fig. 8a). Unfortunately, we are not sure why tRNA-Ile and tRNA-Ala are special and wonder if it could be by chance. The chromosomal tRNA-Ala of *A. tunicatorum* seems to have been acquired by horizontal gene transfer because it was partially aligned with tRNA-Ala in the *rrn* plasmid (Identity 86.96%, Query

coverage 30%) and the top hit of blastn searches versus nr was that of *Maribellus comscasis* WC007 (CP046401.1, Identity 79.69%, Query coverage 87%). Note that nr contains the complete genome of *Persicobacter* sp. JZB09.

Line 355. E-value of 1e-5. What identity and length threshold is this (approximately) and how sensitive are findings to varying this parameter?

Response (R1C11): We set no threshold for the identity and length here, because this step was pre-screening and we prioritized sensitivity. For example, the reference 20 used e-value <10⁻⁵, identity >50%, and query coverage >0.5 but these thresholds missed *rrn* plasmid of *A. ureilytica* (Rep of pAU20*rrn* had a 38.03% identity to CAA71788.1). Likewise, Rep genes of *Treponema sacchrophilum* and *Persicobacteraceae* had the lowest identity 26.25%.

Line 698. 94% is a peculiar threshold. What is the rationale?

Response (R1C12): We agree that 94% seems peculiar, but Retime CIs adopt 94% thresholds (doi.org/10.1093/molbev/msz236). We have revised the legends of Fig. 2 and Extended Data Fig. 2 and added the reference (ref 85).

Table 1. Are these assemblies complete (use of 'scaffold' suggests they may be draft genomes)?

Response (R1C13): The assembly levels were complete except for *P. diffluens*. As shown in LINES 406-412 of Materials and Methods, we confirmed that its chromosome, some plasmids, and the *rrn* plasmid were circular using Unicycler and Bandage, and its result of BUSCO was comparable to the complete genomes in *Persicobacteraceae*.

Ext. dat. Fig. 2. has non-prokaryotes on the tree (genus *Drosophila*, *Athalia*), presumably because they were the source for candidate species, but this should be explained.

Response (R1C14): Thank you for pointing this out. They are presumably contaminations. We have removed non-bacterial sequences and reperformed phylogenetic analysis with the remaining 2,816 sequences. Accordingly, main text (LINES 439-442), Extended Data Fig. 3, and Supplementary Information 6 have been changed.

Minor/stylistic/typographical suggestions:

Line 52. Vague — are you referring to the plasmid genomes, or the bacterial genome as a whole?

Response (R1C15): Thank you for pointing this out. We have changed “genomes” to “bacterial genomes” in the abstract (LINE 47).

Line 61. “Potential of encoding” unclear. Suggested rewrite: “by the fact that chromosomes encode essential genes”.

Response (R1C16): Thank you for your suggestion. We have accordingly revised the sentence (LINES 56-57).

Line 62. How to determine the ‘most’ essential? Are these more important than e.g. DNA polymerase? Reference required.

Response (R1C17): We have changed “most” to “representative” (LINE 57).

Line 71. Is reference 13 relevant here?

Response (R1C18): We apologize for citing a wrong paper. We have now revised the reference 13.

Line 103. How did you *confirm* that they didn’t make false positives or false negatives?

Response (R1C19): We confirmed this by using complete genome datasets of *Aureimonas* complete genomes as written in the text (LINES 98-100).

Paragraphs starting line 111ff. Make it clear here that these paragraphs are referring to the Genbank genome references.

Response (R1C20): We have accordingly referred to the Genbank genome references (LINES 107-108, 114).

Line 201. Or they have more opportunity to acquire such genes?

Response (R1C21): We did not see a tendency that plasmids with abundant Rep families acquire *rrn* operons from Fig. 3a. Thus, we assume that opportunities alone do not explain the specialty of Rep_3-family.

Line 223 onwards. Copy number can be variable though — these counts are from cultures grown specifically for DNA extraction, which might favour different numbers from those in the natural habitat, which should be acknowledged.

Response (R1C22): We have added what you pointed out to the discussion (LINES 348-350).

Fig 1e. Does black indicate no match? White would be more intuitive.

Response (R1C23): Yes, black indicates no match. We have accordingly changed the background color to white in Fig. 1e.

Fig 3c. A dotted line for CN=1 would help orientate readers.

Response (R1C24): We have accordingly added a dotted line to Fig. 3c.

Fig. 4. Are the data presented here corrected for plasmid copy number?

Response (R1C25): No, they are not. For example, the copy number of the tRNA-

Ile gene (anticodon GAT) of *T. saccharophilum* was counted as 1 in Fig. 4 although the effective copy number of the tRNA-Ile gene was 12.

Line 247. Not clear. Do *A. ureilytica* and *O. saccharovorans* have plasmid partition systems or not?

Response (R1C26): No, they don't have the systems. We have accordingly revised the sentence (**LINE 253**).

Data set — typographical error ('withough' should be 'without').

Response (R1C27): We apologize for the error. The error has been corrected.

Reviewer #2 (Remarks to the Author):

In this manuscript, Anda and colleagues systematically survey the NCBI RefSeq database to uncover novel bacterial clades that carry their *rrn* operons exclusively on plasmids. This is interesting because it challenges the assumption that essential genes should be carried preferentially on the chromosome to ensure a stable inheritance. First, the authors uncover four new bacterial clades that add to the two previously known clades (*Aureimonas* spp and *Oecophyllibacter* spp, refs 11 & 18). Then they demonstrate that *rrn*-plasmid associations are stable over evolutionary timescales. They also show how *rrn*-Rep_3 plasmids and their host chromosomes have co-evolved, particularly by increasing the number of available tRNA genes.

I am not a bioinformatics expert, so my review focuses on the biological significance of the results, particularly regarding plasmid biology. The results are potentially interesting, but I had trouble understanding some parts of the manuscript, and some decisions regarding data analyses seem arbitrary. Perhaps a more detailed methods section would help in justifying/explaining the author's

choices (see below for some specific examples). Some sentences are also hard to understand because of language issues.

Response: The manuscript was edited by a native speaker of English from a company providing professional English editing service (Editage).

Specific comments

The criteria for finding *rrn*-containing plasmids warrants a more detailed explanation. For instance, why <35 kb? There are plasmids certainly larger than that. Furthermore, after reading the methods, it is unclear how the presence of ‘full-length rRNA genes’ is determined and why contigs containing single-copy essential genes were discarded (also, what defines a single-copy essential gene?)

Response (R2C1): Thank you for your constructive comments. We have provided details of criteria (LINES 96-98, 386-388), added Extended Data Fig. 1 to the manuscript. Because *rep* genes can also be coded in chromosomes, the criterion that *rrn* operon is present only in contigs with a *rep* gene was not enough (Extended Data Fig. 1a). Thus, to distinguish between chromosomes and plasmids in draft genomes, we used ad-hoc thresholds on contig lengths and numbers of single-copy essential genes (Extended Data Fig. 1b). The single copy essential genes were based on bac120 in the GTDB classification²⁴ and identified by GTDBtk²⁵. To set thresholds, we referred to plasmid sizes of *A. ureilytica*, *Treponema saccharophilum*, and *Persicobacter* spp. These thresholds were for efficient pre-screening purpose and may miss some bacteria that contain *rrn* only on plasmids. We expect that loosening the thresholds may find more *rrn*-containing plasmids in future studies. The definition of full-length rRNA genes followed the annotations (partial-length rRNA genes are marked by “<” or “>” in RefSeq.

I wonder how using specialized software to identify plasmid sequences, such as plasmidfinder (PMID: 31584170) or mob-suite (mob-typer; PMID: 30052170), would affect the author's results.

Response (R2C2): Specialized software assumes that *rrn* plasmids are unlikely and not appropriate for this study. In addition, Plasmidfinder cannot identify *rrn* plasmids of *A. ureilytica* as plasmids because *A. ureilytica* is not a target taxon of plasmidfinder. Mob-suite cannot detect any rep_type(s) from *rrn* plasmid of *A. ureilytica* (T).

In certain bacterial species, it is not uncommon to find plasmids with multiple replicons. It would be interesting to know if the Rep_3 *rrn* plasmids carry a single or multiple replicons

Response (R2C3): Thank you for your constructive comment. All strains of the genus *Persicobacter* and some of *A. ureilytica* (NBRC 106439^T and NS364) have Rep_3 *rrn* plasmids carrying multiple replicons (Fig. 1ace, green-colored genes, Extended. Data. Fig. 3).

The discussion section is vague, with some paragraphs being extremely speculative. It may be better to include a thorough discussion of the limitations of the work.

Response (R2C4): Thank you for pointing it out. We have added a discussion regarding the limitations of this work (**LINES 345-355**)

I need help understanding the last sentence of the abstract (L54). Please re-write. *Such bacteria lacking chromosomal rrn operons have potential for examining and expanding the true potential of plasmids in microbiology and microbial engineering.*

Response (R2C5): We have revised the sentence as follows (**LINES 49-50**).

Such bacteria lacking chromosomal rrn operons have potential for examining and expanding the true potential of plasmids in microbiology and microbial engineering.

→ *Such bacteria lacking chromosomal rrn operons expand our fundamental understanding and microbial engineering of plasmids.*

Please revise the taxonomic nomenclature according to recent changes (e.g. Proteobacteria = Pseudomonadota)

Response (R2C6): Thank you for pointing this out. We have revised Fig. 1, Supplementary Information 6, and the manuscript (LINES 44, 133).

L145. Since rrn operons exclusively located on plasmids were already discovered in two species (refs 11 and 18), this sentence seems an overstatement. Tone down or remove.

Response (R2C7): In this study, we discovered for the first time a clade whose genome organization is conserved at the family level (refs 11 and 18 at the "species level"), so this is not an overstatement.

Supp. Table. 'withough'

Response (R2C8): We apologize for the error. We have corrected Supplementary Information 4 & 6.

Figure 2 legend: 94% confidence? It is more common to plot 95% IC. Is there a reason to choose 94 instead?

Response (R2C9): Retime CIs adopt 94% thresholds (doi.org/10.1093/molbev/msz236). We have revised the legends of Fig. 2 and Extended Data Fig. 2 and added the reference (ref 85).

The term steal is repeatedly used throughout the article. However, a less anthropomorphic term would be more appropriate. Transfer?

Response (R2C10): We changed the term "steal" to "transfer" throughout the manuscript (LINES 68-69, 168).

L187: How a related species is defined? Figure 2f: I am unsure that the term 'copy number' is appropriate here.

Response (R2C11): Thank you for your comment. The related species were defined as same as in the divergence time estimation (LINES 453-455), that is, phylogenetically closest clades carrying chromosomal *rrn* operons.

We agree that multiple Rep_3 genes in a genome are not necessarily a copy. We have removed "Copy" in Fig. 2f.

L191: I do not see how the authors reached that conclusion. From what I see, several strains of the Persicobacteraceae carry only Rep-3 plasmids, but that does not mean that 'Rep_3-family genes were intracellularly transferred and expanded among plasmids' The authors should better explain this claim

Response (R2C12): In Ex. Data. Fig. 3ce, open stars indicate Rep_3 family genes that were intracellularly transferred between the *rrn* plasmid and plasmid. We have added an explanation to the manuscript (LINES 190-193).

L222: Please refer here to Extended data figure 3

Response (R2C13): We have accordingly referred to Extended Data Fig. 4 (previously Extended Data Figure 3 (LINE 230)).

L344: lumen?

Response (R2C14): We apologize for the mistake. We have revised the manuscript (LINE 371).

Reviewer #3 (Remarks to the Author):

The authors were initially inspired by reports of sole *rrn*-operons on plasmids in *A. ureilytica* and *O. saccharovorans*, and subsequently investigated the presence of

this phenomenon in *Persicobacteraceae*. They identified six strains from four genera that harbored plasmid-borne *rrn*-operons, potentially as a result of a translocation event from the chromosome. The plasmid-borne *rrn*-operons were found to have a higher copy number compared to those on the chromosome, which corresponded to an increase in the number of tRNA gene clusters on the chromosomes.

1. They found three additional genomes has sole *rrn* operons in plasmids out of 86,822 genomes (0.003%). This result means such case is really rare, at least based on the dataset they used.

Persicobacter spp (related *Persicobacter diffluens*); *Treponema saccharophilum*

2. Based on the bioinformatic analysis, they further explored *P. diffluens* relatives: *P. psychrovividus*, *Aureibacter tunicatorum*, and *Fulvitalea axinellae*. In total they found six strains in such case and confirmed two strains (*P. diffluens* NBRC 15940T and *T. saccharophilum* JCM 32279T) by sequencing. Turn out *Persicobacteraceae* all in such case.

Taken together, this manuscript reports an interesting 'exception to the rule' on the evolution of plasmid gene content.

Comments according to their order in the manuscript:

Lines 66, 167. The citation of refs 9,10 as contradicting the findings of *rrn* on plasmids is not entirely justified. I suggest that the authors read those publications carefully. In essence, these studies explain why essential genes in plasmids should be rare, which is exactly what this study shows – *rrn* genes were found in 0.003% of all analyzed genomes. I would argue that this study is rather in agreement with both refs 9 & 10 (but for slightly different reasons since ref 10 at

least deals with protein coding genes and dose effect, which is less relevant for RNA genes). Line 165-167: “It does challenge the belief that essential gene cannot be maintained stably on plasmids.” The current state is, a second copy of essential genes cannot be maintained stably on plasmids, sole essential gene can be maintained on plasmids. Again, please read carefully refs 9,10.

Response (R3C1): Thank you for this insightful comment. First, we noticed that the citation of reference 10 was incorrect and have accordingly corrected it. Reference 9 shows that bacteria that exclusively encode essential genes on plasmids should be very rare, which is consistent with our finding in this study as suggested by reviewer. What challenges the belief is that such a genomic structure cannot be maintained for an evolutionarily long term according to the population dynamics model. We have accordingly revised the manuscript (**LINES 60-62, 165-166**).

Line 102: They focus only on small plasmids containing *rrn* operons. (Contig size <35 kb) How does this cut-off value come up?

Response (R3C2): To distinguish between chromosomes and plasmids in draft genomes, we used an ad-hoc threshold on contig lengths (**Extended Data Fig. 1b**). We referred to plasmid sizes of *A. ureilytica*, *Treponema saccharophilum*, and *Persicobacter* spp. to set the 35 kb thresholds. This threshold was for efficient pre-screening purposes and may miss some bacteria that contain *rrn* only on plasmids. We expect that loosening the threshold may find more *rrn*-containing plasmids in future studies.

Line 675 (Figure 1): I checked the genome of *Persicobacter* sp. JZB09, the authors reported here the plasmid of size 30,222 bp is actually the smallest plasmid out of 16 plasmids in the same strain, this plasmid is also named JZB09-Plasmid16. In RefSeq the accession number of this contig is NZ_CP012859.1, which has been removed later by NCBI staff. From our experience such ‘removal’ of plasmids may

occur occasionally and these correspond (most likely) to tackling of in assembly error. In other words, my interpretation would be that this contig was erroneously annotated as a plasmid to begin with and is actually part of the chromosome.

Please confirm the state of this assembly.

Response (R3C3): We also verified that this genome was removed by NCBI staff and asked NCBI the reason by email. While we have not received a reply, the notation itself was then removed. We suppose that the assembly status is now fixed.

Line 675 (Figure 1): I noticed the synteny of *rrn* operons in Figure 1e, would it be possible using more rRNA genes to build the tree in Figure 1f instead of only 16S rRNA gene? Common structure: *rrs* (16S rRNA)-*trnI* (tRNA^{Ile})-*trnA* (tRNA^{Ala})-*rrl* (23S rRNA)-*rrf* (5S rRNA)

Response (R3C4): We have performed phylogenetic analysis using the 16S, 23S, and 5S rRNA genes. Details are explained with R3C5.

Line 149-156 The similarity between the topologies of core genes and 16s rRNA should be tested statistically. This can be done, e.g., with IQtree (using the core genes topology as a user tree for the 16 rRNA alignments and performing a likelihood test).

Response (R3C5): We have performed phylogenetic analysis using the concatenated 16S, 23S, and 5S rRNA genes, by collecting additional sequences when our dataset contained partial sequences only (Supplementary Information 1, 2). Fig. 1fg have been accordingly replaced and the original one based on 16S has been moved to Extended Data Fig. 5. The rRNA phylogenetic tree was again compared with that of core genes, and no difference in topology was observed (Supplementary Information 3). The main text has been revised (LINES 149-153, 421-431).

Line 148: Admittedly, I am not a fan of timed phylogenies for ancient splits in prokaryotes. I prefer not to comment extensively on this part. I think that it is sufficient to state that, e.g., the plasmid acquisition occurred before/after a specific species divergence event. I also note that the dating error bars in Fig. 2a-d span many ancestral nodes so I would not base any conclusion based on that inference alone. I could only recommend to read the publications of those dating methods carefully in order to better understand their assumptions and caveats.

Response (R3C6): We agree with you that there are substantial debates on the accuracy of divergence time estimation in bacteria; however we also note that such estimates have been useful to the general audience. For example, bacterial divergence time was used to estimate dynamics of lateral gene transfer (doi.org/10.1128/mbio.00644-17). Here, we have accordingly added a note on the limitation of our analysis (**LINES 322-323**).

Line 169: “stole”? I cannot imagine how did these small plasmids stole *rrn* operon from chromosome, there is no protein related to any MGEs on the same plasmid; through recombination? I recommend using terminology from the field of molecular evolution here – that is – either the operon was translocated from the chromosome to the plasmid, or an alternative copy was acquired (as suggested in Line 187 if I get it right) and subsequently the chromosomal operon was lost.

Response (R3C7): We have changed the word ‘steal’ to “transfer’ throughout the manuscript (**LINES 68-69, 168**).

Line 187 & line 754: Extended Data Fig. 2. I don’t understand how were the transfer events inferred? Marked with stars. And Plasmid pJZB16 is a *rrn*-plasmid, isn’t it? Why it has been marked as the other plasmid with an empty triangle? Could the authors supply the sequences, alignment and tree file for the phylogenetic trees in extended data Fig. 2?

Response (R3C8): Thank you for pointing them out. We inferred the transfer events using phylogenetic analysis and synteny analysis (Extended Data Fig. 3, Fig. 1e). For example, *P. diffluens* JZB09 does not have *repA1* on the *rrn* plasmid but on JZB09-Plasmid11 (Fig. 1e and Extended Data Fig. 3c) although the other two strains of *P. diffluens* have *repA1* on the *rrn* plasmids. These results suggested that *repA1* was transferred between the *rrn* plasmid and (non-*rrn*) plasmid. We apologize for the error that the pJZB16 was plasmid JZ09-Plasmid 15 without *rrn* operon and this has been corrected (Supplementary Information 6). We have added an explanation (LINES 190-193) and provided the sequences, alignment, and tree files as Supplementary Data.

Line 188: 'Holders' -> better use 'Hosts'

Response (R3C9): Thank you for your suggestion. We have accordingly revised (LINE 187).

Line 191-193: How did a replication initiation protein transfer between plasmid intracellularly? Based on the phylogenetic tree in extended Data Fig. 2?

Response (R3C10): Please refer to our answer to your comment (R3C8).

Line 196-198: "we hypothesized that Rep_3-family plasmids would have a propensity to carry rRNA genes." According to the phylogenetic tree in extended Data Fig. 2, I don't see any clue for this hypothesis. Also – Fig. 3a suggests that the authors performed a test to compare the presence of rRNA gene among all Rep types (i.e., a 2xn contingency table) while the more suitable analysis to test this hypothesis would be with a 2x2 design – i.e., Rep_3-family versus all the rest.

Response (R3C11): Thank you for pointing them out. The phylogenetic tree of Extended Data Fig. 3 does not carry any replicon information or co-occurrence information with *rrn* operons derived from bacteria with chromosomal *rrn* operons. Even if we map such information to the phylogenetic tree, we do not consider that

we can arrive at this hypothesis because there are very few plasmids of the Rep_3 family with rRNA genes (Fig. 3a). We tested 2x2; Fig. 3ab shows the other Rep families so that we can see the breakdown.

Line 195-208: This part is very controversial, what is the special character of Rep_3-family? I tend to believe it's not only due to the plasmid but also the host. It's a neutral event, not a destiny that Rep_3-family plasmids will acquire any essential genes from chromosomes for sure. The following test for presence of core gene is not clear to me.

Response (R3C12): Thank you for your comment. While we do not have any strong hypothesis on this issue, we note that Rep_3 family genes are detected from iteron plasmids. Iteron plasmids control their copy numbers by handcuffing (doi.org/10.1046/j.1365-2958.2000.01986.x) and possibly localize in the cell. They might have nucleolus-like structures (doi.org/10.3389/fmicb.2018.01115) or special structures for prevent transcription and translation crushes (doi.org/10.1146/annurev-micro-090817-062514). To investigate if the host has any characteristics behind the losses of chromosomal *rnm* operons, we performed ancestral state estimation of gene composition (Fig. 2e). We did not identify any KO, Tigrfam, or Pfam annotations that were common among the four clades, except for Rep_3 (Fig. 2e). We also agree that the Rep_3-family plasmid may have acquired essential genes from chromosomes randomly. We have accordingly revised the manuscript (**LINES 205, 215-216**), also by adding an explanation of a test for presence of core gene (**LINES 209-210**).

Line 210 - 230: Estimating the relative plasmid copy number from short read sequencing is legit (and it would be good to write here something about the comparison with ref. 11). The assessment of chromosome copy number (or ploidy) is very problematic here – I don't see anything in the methods that can explain how

these number were obtained. The part about the plasmid high copy number being advantageous for having the *rrn* on a plasmid should be in the discussion.

Response (R3C13): We have added a sentence on comparison with ref. 11 (LINES 222-225). We did not estimate chromosome copy numbers, but estimated relative numbers. We have added an explanation how these numbers were obtained (LINES 488-492) in the materials and methods. We have added a part about the high copy number of *rrn* plasmids in the discussion (LINES 350-351, 372-377).

Line 212: “Rep_3-family genes may play a role in controlling plasmid copy numbers” – Better place in the discussion part.

Response (R3C14): Thank you for the advice, but this sentence is needed to here to give rationales for the succeeding analysis.

Line 227-228: “High-copy number plasmids can be inherited stably even by stochastic segregations.” – this assumption (the authors did not test the stochastic segregation/passive partition) aligns with the common view in the field and would be still good to have here a citation.

Response (R3C15): We have cited two references (refs 31, 32) to support the explanation.

Line 237: “In total, the copy numbers of *rrn* operons per cell remained constant (Fig. 3d)” – Fig.3d shows something else which I’m not sure how is related to this sentence.

Response (R3C16): ‘The copy numbers of *rrn* operons per cell’ is equivalent to ‘Effective copy number of *rrn* operons’ or ‘gene dosage’. For example, *Persicobacter diffluens* has about 4 copies of the *rrn* plasmid (Fig. 3c). Since there are 3 copies of *rrn* operons on the plasmid (Fig. 1a), the effective copy number of *rrn* operons are 12 (Fig. 3d). Similarly, *Treponema saccharophilum* has 12 copies

of *rrn* plasmid, and there is only one copy of *rrn* operon on the plasmid, thus the effective copy number of *rrn* operons are 12 (Fig. 3d). These results lead to the interpretation in LINES 244-245.

Line 238-239: “To our knowledge, no previous studies have directly observed that evolutionary pressure actively maintained copy numbers of plasmids and *rrn* operons.” – I am not sure that the authors show here a direct observation for that statement either. That is – they do not test the effect of evolutionary pressure on the plasmid copy number.

Response (R3C17): According to the comment, we have removed ‘directly’ (LINE 246).

Line 244-245: “which was likely the first empirical confirmation of a prediction that a plasmid obtains a partition system after acquisition of essential genes” – I don’t think that the authors showed empirically that the acquisition of the *parAB* occurred after the *rrn* acquisition. This may be ‘inference’ from phylogenetic trees (and the such) and should be communicated as such.

Response (R3C18): The inference is based on phylogenetic analysis, so the text has been changed (LINE 258).

Line 248-252: better move this to the discussion.

Response (R3C19): We have moved the above sentence to the discussion (LINES 361-365).

Line 268-270: again – better to move to the discussion. This is not a clear result and is somewhat speculative.

Response (R3C20): We have moved the above sentence to the discussion (LINES 353-355).

Line 254: The last section on tRNA is also not very convincing in the absence of a 'contrast' to compare those observations to a group without plasmid-encoded *rrn*. This maybe as well be a Persicobacteraceae specific phenomena. The history and the mechanism were not clear to me.

Response (R3C21): We have added a part on tRNA to the discussion (LINES 350-355).

Lines 299-304 and Fig. 5 – Im not sure that I have seen clear evidence for a traslocation of the chromosomal *rrn* to the plasmid rather than an acquisition of a 'foreign' *rrn* and loss of the chromosomal copy.

Response (R3C22): Although we did not see a clear discrepancy in the topology of the phylogenetic trees based rRNA genes and core genes, but it may still be possible that a foreign *rrn* was transferred from a closely related strain. We have revised the sentence accordingly (LINES 152-153).

Line 313-314 – considering the parent of gene presence/absence, I would argue that the loss of the chromosomal copy is the most parsimonious event, without invoking arguments on selection pressure in the past (which are unknown).

Response (R3C23): We actually agree with you that the loss of the chromosomal copy is the most parsimonious event. However, we keep this another hypothesis for thorough discussion.

The discussion overall includes some almost 'off topic' subjects like Lines 315-322. Including the topics discussed in the results section will make it more interesting and to the point.

Response (R3C24): Thank you for your advice. We have accordingly moved the topics in the Results section to Discussion; Lines 325-333 (previously 315-322) were left as they describe the limitations of this study as pointed out by Reviewer 2 (R2C4). The MAG and SAG would be important future directions based on this study, according to a recent paper on the construction of *A. ureilytica* circular

chromosome without chromosomal *rrn* operons. We have accordingly revised the manuscript (LINES 334-335).

The last discussion paragraph is quite speculative. I would say that having *rrn* only on a plasmid is a 'super addiction mechanism' that requires no further explanations for the plasmid stability in the host.

Response (R3C25): We agree that this paragraph is speculative, but it should be discussed that *rrn* operon can be back to the chromosome.

Reviewer #1 (Remarks to the Author):

The authors have addressed my main concerns satisfactorily, and only minor comments/typos/suggestions were apparent when reading the revised manuscript.

Line 57. Suggest replace 'representative' with 'canonical'.

Line 60: 'extrachromosomes' should be clarified as 'extrachromosomal replicons'.

Line 107ff. I noticed that at least one of the genomes from NCBI analysed by the authors (Persicobacter JZB09) is not associated with a publication. Though it was deposited ~8y ago and is publicly available (not covered by embargo as far as I am aware), I wonder if the authors know the intention of the original submitter and/or if they have been in contact. I am aware of problems arising from re-analysis of unpublished but publicly available genome sequencing in the past <https://www.science.org/doi/10.1126/sciadv.abh1051>

Line 126. It would be helpful to include a short sentence stating clearly that neither of these genomes had chromosomal rrn.

Line 210. It is not clear what 'their' is referring to, the genes or the plasmids. Rephrase.

Furthermore, 'degradation' should (probably) be replaced with 'segregation' or 'loss'.

Line 225. It is also possible that discrepancies in culture technique cause variation in plasmid copy number.

Line 272. This *may* be because — the authors do not actually test this hypothesis. (It is plausible for a genome to generally reduce in size but individual genes to increase or retain copy number).

Line 287/288. A short sentence on the implication of this observation would be helpful.

Line 318. Things are either unique or they are not. Suggest deletion of word 'quite' (also avoids misunderstanding between British and American English use of 'quite').

Fig. 1e shows nucleic acid comparisons, but the scale is in AA% identity (possibly because the comparisons have been done with tblastx?) — this should be explained.

Line 790. I think the authors have this the wrong way round in the legend — 'presence' is on the right, not the left. Also line 792.

Fig S8d. What is the scale in the colour matches, is it the same as 1e? Indicate (and consider changing the background colour to be consistent).

Reviewer #2 (Remarks to the Author):

The authors have answered all my comments satisfactorily. I believe the new version has greatly improved and it's easier to understand. I congratulate the authors for this interesting work!

Reviewer #3 (Remarks to the Author):

The authors responded to most of my comments quite convincingly.

Im still not entirely convinced by their evidence for rrn transfer from the chromosome to the plasmid, but I dont see a major flaw in their arguments (and its their opinion). Nonetheless, I would like to draw the authors attention to the data they used for the reconstruction of Rep_3 phylogeny. I viewed the alignment that they now supply (Ex_Dat_Fig_3_Rep3_alignment.faa; with the online tool MView from EBI) and it seems to contain some sequences that are significantly shorter compared to the 'typical' length (e.g., WP_026708947.1, WP_052700784.1) and also some that have generally extremely low sequence similarity overall (e.g., <20% identical amino acids or even 8% (!), e.g., WP_060577704.1, PEDI_53320). Including such sequences in the alignment and phylogeny may lead to artifacts and biased conclusions.

Reviewer #1 (Remarks to the Author):

The authors have addressed my main concerns satisfactorily, and only minor comments/typos/suggestions were apparent when reading the revised manuscript. We are honored that you are satisfied with our revisions. Thank you for your constructive suggestions.

Line 57. Suggest replace 'representative' with 'canonical'.
We have replaced 'representative' with 'canonical'. (LINE 44)

Line 60: 'extrachromosomes' should be clarified as 'extrachromosomal replicons'.
We have clarified 'extrachromosomes' as 'extrachromosomal replicons'. (LINE 47)

Line 107ff. I noticed that at least one of the genomes from NCBI analysed by the authors (Persicobacter JZB09) is not associated with a publication. Though it was deposited ~8y ago and is publicly available (not covered by embargo as far as I am aware), I wonder if the authors know the intention of the original submitter and/or if they have been in contact. I am aware of problems arising from re-analysis of unpublished but publicly available genome sequencing in the past <https://www.science.org/doi/10.1126/sciadv.abh1051>
<<https://www.science.org/doi/10.1126/sciadv.abh1051>>
We thank the reviewer for carefully and kindly pointing out this issue. We have not contacted the original submitter because the JZB09 genome is included in NCBI RefSeq, which provides rapid, unrestricted public access to submitted data unless data are requested to be held until a specified date, or the publication date, whichever comes first (JGI statement: <https://jgi.doe.gov/statement-on-the-use-of-genomics-data/>). Furthermore, the genome does not fall under the JGI Data policy. We also checked other genomes that are not associated with publications and confirmed that the JGI Data Policy did not cover them.

Line 126. It would be helpful to include a short sentence stating clearly that neither of these genomes had chromosomal *rrn*.
We have added a sentence 'Neither of those genomes had *rrn* operons in their chromosomes.' (LINE 110)

Line 210. It is not clear what 'their' is referring to, the genes or the plasmids. Rephrase. Furthermore, 'degradation' should (probably) be replaced with 'segregation' or 'loss'. Thank you for pointing them out. We have rewritten them as follows to make the meaning clearer.

This was because, for example, the Rep_3-family genes could make plasmids extremely stable over generations by decreasing their degradation rates⁹, so that essential genes could have been maintained on plasmids.

→This was because, for example, the Rep_3-family genes could make plasmids extremely stable over generations by decreasing the plasmid loss rates, so that essential genes could have been maintained on plasmids. (LINE 194)

Line 225. It is also possible that discrepancies in culture technique cause variation in plasmid copy number.

Thank you for your suggestion. We have revised the sentence as follows.

This difference could be attributed to different methodologies because the previous study used qPCR (*rrs/rpsB*), which can be affected by a PCR amplification bias.

→ This difference could be attributed to discrepancies in culture techniques or methods to determine plasmid copy numbers. For example, the previous study used qPCR (*rrs/rpsB*), which can be affected by a PCR amplification bias. (LINES 208-209)

Line 272. This **may** be because — the authors do not actually test this hypothesis. (It is plausible for a genome to generally reduce in size but individual genes to increase or retain copy number).

We have replaced ‘would’ by ‘may’. (LINE 257)

Line 287/288. A short sentence on the implication of this observation would be helpful.

We have added the following sentence in LINES 273-275.

Those results suggest that bacteria whose *rrn* operon is only on a plasmid were under selection pressure leading to faster growth rates and increased effective numbers of *rrn* operons and tRNA genes.

Line 318. Things are either unique or they are not. Suggest deletion of word ‘quite’ (also avoids misunderstanding between British and American English use of ‘quite’).

Fig. 1e shows nucleic acid comparisons, but the scale is in AA% identity (possibly because the comparisons have been done with tblastx?) — this should be explained.

Thank you for your suggestion. We have deleted ‘quite’. (LINE 305)

Fig. 1e is based on tBlastx to calculate similarities of CDSs on the plasmid by avoiding performance of CDS prediction. Indeed, we could not detect the synteny of the biotin synthesis genes of *P. diffluens* and *P. psychroviduus* using BLASTN and BLASTP. We are aware that this has the disadvantage that rRNA and tRNA genes are also subject to tBLASTx.

Line 790. I think the authors have this the wrong way round in the legend — ‘presence’ is on the right, not the left. Also line 792.

We apologize for the errors in the legend of Fig. 3ab. The errors have now been corrected.

Fig S8d. What is the scale in the colour matches, is it the same as 1e? Indicate (and consider changing the background colour to be consistent).

Yes, it is the same as Fig. 1e. We have revised it and changed the background color.

Reviewer #2 (Remarks to the Author):

The authors have answered all my comments satisfactorily. I believe the new version has greatly improved and it's easier to understand. I congratulate the authors for this interesting work!

We are honored that Reviewer 2 was satisfied with our revision.

Reviewer #3 (Remarks to the Author):

The authors responded to most of my comments quite convincingly.

Im still not entirely convinced by their evidence for *rrn* transfer from the chromosome to the plasmid, but I dont see a major flaw in their arguments (and its their opinion). Nonetheless, I would like to draw the authors attention to the data they used for the reconstruction of Rep_3 phylogeny. I viewed the alignment that they now supply (Ex_Dat_Fig_3_Rep3_alignment.faa; with the online tool MView from EBI) and it seems to contain some sequences that are significantly shorter compared to the 'typical' length (e.g., WP_026708947.1, WP_052700784.1) and also some that have generally extremely low sequence similarity overall (e.g., <20% identical amino acids or even 8% (!), e.g., WP_060577704.1, PEDI_53320). Including such sequences in the alignment and phylogeny may lead to artifacts and biased conclusions.

We are honored that you are satisfied with our revisions. Meanwhile, thank you for your kind attention to the issue regarding the Rep_3 phylogenetic reconstruction. As you pointed out, the sequences we used contained sequences with low similarity or short sequences. This is because the sequences containing the Rep_3 family of bacteria without chromosomal *rrn* operons themselves have such problems (e.g. PEDI_53320, WP_060577704.1). Since the purpose of this analysis was to determine the phylogenetic position of Rep_3 in bacteria without chromosomal *rrn* operons, low-identity or short sequences were included in the analysis.